# Large-scale state-dependent membrane remodeling by a transporter protein

Wenchang Zhou[1], Giacomo Fiorin[1], Claudio Anselmi[1†],
Hossein Ali Karimi-Varzaneh[2‡], Horacio Poblete[1,2§], Lucy R Forrest[2*],
José D Faraldo-Gómez[1*]

[1]Theoretical Molecular Biophysics Laboratory, National Heart, Lung and Blood Institute, National Institutes of Health, Bethesda, United States; [2]Computational Structural Biology Section, National Institute of Neurological Disorders and Stroke, National Institutes of Health, Bethesda, United States

**Abstract** That channels and transporters can influence the membrane morphology is increasingly recognized. Less appreciated is that the extent and free-energy cost of these deformations likely varies among different functional states of a protein, and thus, that they might contribute significantly to defining its mechanism. We consider the trimeric Na$^+$-aspartate symporter Glt$_{Ph}$, a homolog of an important class of neurotransmitter transporters, whose mechanism entails one of the most drastic structural changes known. Molecular simulations indicate that when the protomers become inward-facing, they cause deep, long-ranged, and yet mutually-independent membrane deformations. Using a novel simulation methodology, we estimate that the free-energy cost of this membrane perturbation is in the order of 6–7 kcal/mol per protomer. Compensating free-energy contributions within the protein or its environment must thus stabilize this inward-facing conformation for the transporter to function. We discuss these striking results in the context of existing experimental observations for this and other transporters.

*For correspondence:
lucy.forrest@nih.gov (LRF);
jose.faraldo@nih.gov (JDF-G)

Present address: †Children's National Hospital, Washington, United States; ‡Continental Reifen Deutschland GmbH, Hannover, Germany; §Centro de Bioinformática y Simulación Molecular, University of Talca, Talca, Chile

## Introduction

Integral membrane proteins can have a marked impact on the morphology of the surrounding bilayer. For example, they might alter its curvature or thickness, or foster the enrichment or depletion of specific types of lipid in their vicinity (*Lee, 2004*; *Andersen and Koeppe, 2007*; *Marsh, 2008*; *Phillips et al., 2009*). These perturbations develop to accommodate the amino-acid composition and specific structural features of the protein surface. For a broad range of systems of interest, however, these features are not static, but vary as the protein carries out its biological activity. Examples include channel proteins and their gating mechanisms, or the conformational cycles resulting in alternating access in active transporters. The protein-lipid interface also changes when membrane proteins form complexes or oligomers. Because most membrane perturbations entail an energetic or entropic cost, it seems reasonable to expect that the bilayer contributes to the thermodynamics and kinetics of these processes, favoring or disfavoring specific structural states. Such interdependence would help explain why changes in lipid bilayer composition, either resulting from natural regulatory mechanisms or induced artificially, can have a determining effect on protein function (*Jensen and Mouritsen, 2004*; *Andersen and Koeppe, 2007*; *Denning and Beckstein, 2013*; *Cordero-Morales and Vásquez, 2018*; *Haselwandter and MacKinnon, 2018*).

Here, we seek to gain insights into this interplay for a class of membrane proteins known to undergo a striking structural transformation, namely secondary-active transporters featuring the so-called elevator mechanism. We specifically focus on the Na$^+$-aspartate symporter Glt$_{Ph}$ from *Pyrococcus horikoshii*, a member and model system of the Excitatory Amino-Acid Transporter (EAAT) family, which includes the human SLC1 neurotransmitter transporters (*Gether et al., 2006*).

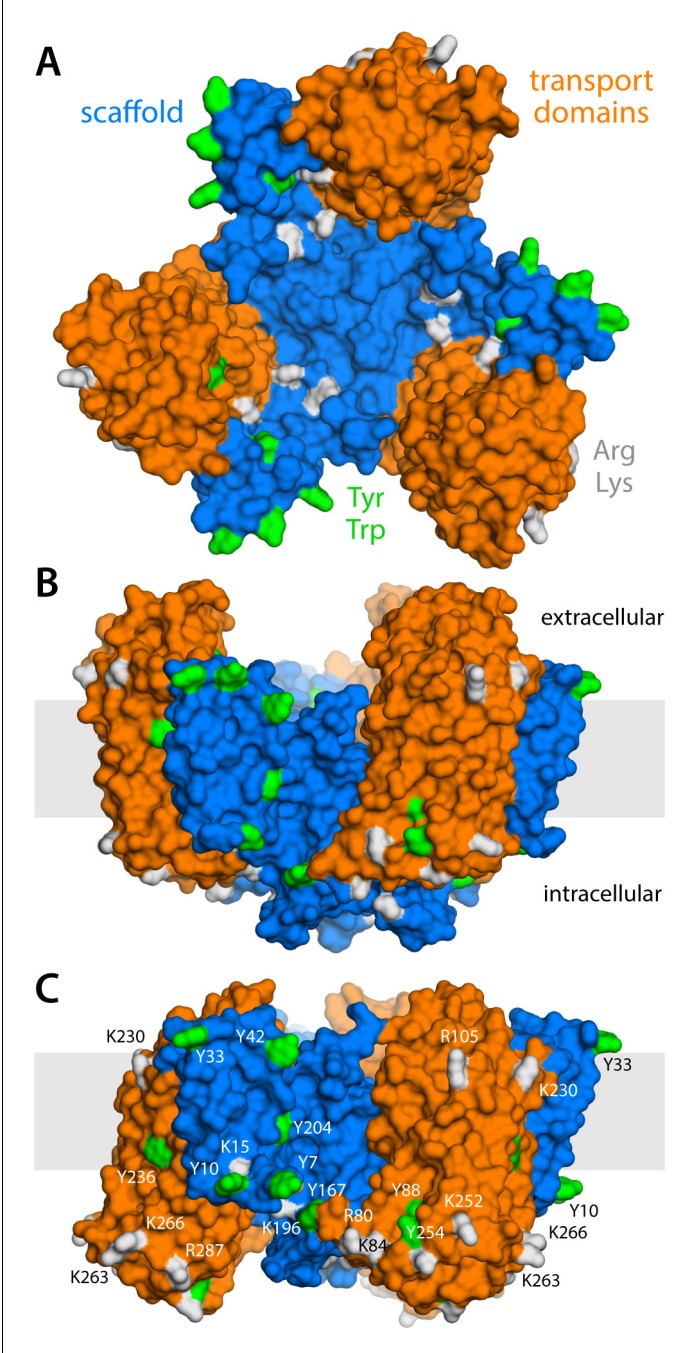

**Figure 1.** Structure of the Glt_Ph trimer in the outward- and inward-facing states. (**A**) View from the extracellular space, with the three protomers in the outward-facing conformation (PDB 2NWL). The 'scaffold' (blue) mediates all protein-protein interactions between protomers. The 'transport' domains (orange) carry the bound substrates, moving relative to the scaffold perpendicularly to the membrane. Side-chains that might form hydrogen-bonds with lipid headgroups are highlighted (Tyr, Trp, Arg, Lys). (**B**) Same as (**A**), viewed along the membrane plane, with the approximate membrane region shown as a gray box. (**C**) Same view as (**B**), with all three protomers in the inward-facing conformation (PDB 3KBC).

The online version of this article includes the following figure supplement(s) for figure 1:

**Figure supplement 1.** Molecular simulation systems.

Secondary-active transporters like Glt$_{Ph}$ interconvert between two major conformational states that alternately expose binding sites for ions and substrate to either side of the membrane (*Figure 1*) (*Jardetzky, 1966*; *Reyes et al., 2009*; *Boudker and Verdon, 2010*; *Forrest et al., 2011*). In Glt$_{Ph}$, this conformational interconversion takes place only when Na$^+$ and aspartate occupy the transporter or when the transporter is empty (*Yernool et al., 2004*; *Boudker et al., 2007*; *Ryan et al., 2009*), which defines this protein as a co-transporter. For either occupancy state, the exchange between outward- and inward-facing conformations is spontaneous, that is it does not require an extrinsic electrochemical driving force. However, the relative populations of these conformational states and the net directionality of the transport cycle do depend on the membrane potential and the relative concentrations of ions and substrates on either side of the membrane. This dependence is why downhill translocation of Na$^+$ will power uphill transport of aspartate, and vice versa.

Glt$_{Ph}$ forms a homotrimer (*Yernool et al., 2004*); yet, each protomer works independently from its neighbors (*Akyuz et al., 2013*; *Erkens et al., 2013*; *Akyuz et al., 2015*; *Ruan et al., 2017*). The protomers consist of two distinct units: the scaffold domain, which provides a stable trimerization interface (*Groeneveld and Slotboom, 2007*; *Verdon and Boudker, 2012*; *Georgieva et al., 2013*; *Hänelt et al., 2013*), and the transport domain, which encapsulates the ion and aspartate binding sites (*Yernool et al., 2004*; *Reyes et al., 2009*). Both domains are exposed to the lipid bilayer, but structures of Glt$_{Ph}$ in outward- and inward-facing states clearly reveal that it is the movement of the latter domain that results in alternating access (*Yernool et al., 2004*; *Reyes et al., 2009*). Specifically, if one assumes that the scaffold domains remain stationary relative to the membrane midplane, each transport domain would move perpendicularly to that plane by approximately 15 Å, that is about one-third of the total bilayer width. How this drastic structural change impacts the relationship between protein and membrane has, to our knowledge, not been previously evaluated. One might envisage that the transport domains simply traverse the membrane (like an elevator), moving polar sidechains on their surface into the bilayer interior and exposing hydrophobic ones to the solvent (*Reyes et al., 2009*). Alternatively, the morphology of the lipid bilayer could adapt to the conformational state of the protein, and match the amino-acid make-up of the protein surface throughout the transport cycle. Which of these two possibilities entails the least energetic cost is entirely unclear. The absence of functional cooperativity among protomers would appear to rule out the latter, as a pronounced membrane deformation induced by one protomer could impact the conformational dynamics of its two neighbors. On the other hand, the cumulative cost of dehydration of polar and charged sidechains could be exceedingly large. Experimental studies of the Glt$_{Ph}$ homolog EAAT2 also appear to show that bilayer-facing regions in the outward-facing state are similarly solvent-accessible throughout the transport cycle (*Silverstein et al., 2013*).

Molecular dynamics simulations in principle provide a means to examine this interplay in great detail (*Cui et al., 2013*; *Marrink et al., 2019*). Current computing power makes it feasible to examine the morphology of simple phospholipid membranes around protein structures, for many cases of interest. Highly complex membranes remain beyond reach, however, if represented in atomic detail, as the relaxation time of an arbitrarily-configured multi-component lipid mixture is slower than typically attainable simulation times, thus imposing a starting-condition bias on any analysis. In such cases, so-called coarse-grained simulations are a viable approach despite their reduced level of detail (*Corradi et al., 2018*). By contrast, simulation methods that permit a direct quantification of the energetic footprint of a membrane morphological change have been lacking. Computational approaches in this area have typically relied on continuum-mechanics theories of membrane elasticity, which necessarily pre-suppose important parameters defining the bilayer energetics (*Argudo et al., 2017*). Although such approaches can be predictive and insightful in some cases (*Mondal et al., 2011*; *Bethel and Grabe, 2016*), it is unclear whether macroscopic models are generally transferable to the length-scales of individual membrane proteins (*Lee et al., 2013*; *Fiorin et al., 2019*). To circumvent this problem, we have recently developed a free-energy simulation method (*Fiorin et al., 2019*) with which the potential-of-mean-force of a membrane deformation can be probed directly from a simulation system, much in the same way processes such as ligand recognition or ion permeation are commonly characterized. Here, in addition to conventional simulations and structural bioinformatic approaches, we apply this new technology to obtain quantitative insights into the nature of the interplay between Glt$_{Ph}$ and the surrounding membrane. The implications of the major conformational change required for transport are discussed in the light of available experimental observations.

## Results

### Alternating access causes major membrane deformations, long-ranged but non-cooperative

To examine how the transport cycle of Glt$_{Ph}$ impacts the surrounding membrane, we carried out molecular dynamics simulations of phospholipid bilayers containing structures of Glt$_{Ph}$ trimers in four different conformational states. Specifically, we examined states where all three protomers are in either the inward- or outward-facing states (*Figure 1*), as well as two intermediates where one protomer is inward-facing and the other two are outward-facing, or vice versa. To enable the calculations to reveal long-range perturbations, the simulated membranes are about $50 \times 50$ nm$^2$ in size, that is six-fold wider than the transporter (*Figure 1—figure supplement 1*). This area translates into ~7500 lipid molecules, much larger than typical simulation systems. In a first set of calculations, therefore, we opted for coarse-grained representation of the molecular system, using the MARTINI forcefield (*Marrink et al., 2007*; *Monticelli et al., 2008*). The conformational state of the trimer was preserved throughout each of the simulations, while the lipid bilayer was free to adjust to that state. It is worth noting that the elastic properties of lipid bilayers, as quantified by, for example, their macroscopic bending modulus, are well described by MARTINI, despite its inherent approximations (*Marrink et al., 2004*; *Fiorin et al., 2019*).

The results of this analysis are summarized in *Figure 2*. For the state with all three protomers in the outward-facing conformation, we found the membrane to be only modestly perturbed. This perturbation is maximal in the vicinity of the transport domains, where the membrane mid-plane is elevated by 3–4 Å; near the scaffold, however, the elevation is only ~2 Å. By contrast, for the cases in which one, two or all three transport domains are in the inward-facing state, we observe a major morphological impact on the surrounding membrane. Specifically, we observe that the displacement of the transport domain causes a ~ 10 Å depression in the lipid bilayer, that is, about 50% of the width of its hydrophobic core. This perturbation is also remarkably long-ranged, extending radially for nearly 100 Å from the protein surface (*Figure 2D*). Strikingly, however, along the perimeter of the protein, the perturbation is entirely confined to the interface with the transport domain, that is, it is abruptly terminated near the scaffold, where the membrane is again minimally depressed, by only about 1–2 Å. Indeed, comparison of intermediates with one or two protomers in the inward-facing state shows that each transport domain causes an independent perturbation, largely identical and seemingly additive to that caused by the other protomers, while in the vicinity of the scaffold domains the membrane remains largely unperturbed.

The data presented in *Figure 2* were obtained for a POPC membrane under no tension (Materials and methods). Similar results were obtained in simulations with applied membrane tensions as high as 10 mN/m (*Figure 3*), and in simulations with bilayers of different composition, namely either POPE, a 2:1 mixture of POPE and POPG (both at 298 K), and DPPC (at 323 K) (*Figure 4*). It should also be noted that the observed deflections in the membrane mid-plane do not result from significant differences in lipid-bilayer thickness (*Figure 4—figure supplement 1*). Analysis of the second-rank order parameter of the lipid alkyl tails, which is a measure of their tilt relative to the membrane perpendicular, also shows little contrast between the outward- and inward-facing conformational states (*Figure 4—figure supplement 2*). The only significant difference is observed for the inner leaflet, at the point where the transport domain meets the scaffold, in the inward-facing state. Here, the lipids become significantly tilted, on average, seemingly to adapt to the abrupt changes in membrane curvature induced by the transport domains. This effect is, however, localized and does not propagate beyond the perimeter of the scaffold. We conclude, therefore, that the long-ranged deformations induced by Glt$_{Ph}$ are most accurately described as bending, rather than other types of perturbation.

Finally, it is interesting to note that our simulations show that the trimerization domain is not completely static relative to the membrane throughout the alternating-access cycle; as indicated in *Figure 2D*, we observe that the scaffold shifts by about 2 Å when the all-outward and all-inward states are compared. Consequently, the vertical displacement of the transport domains relative to the membrane is slightly larger than the ~15 Å that would be deduced from an overlay of the structures.

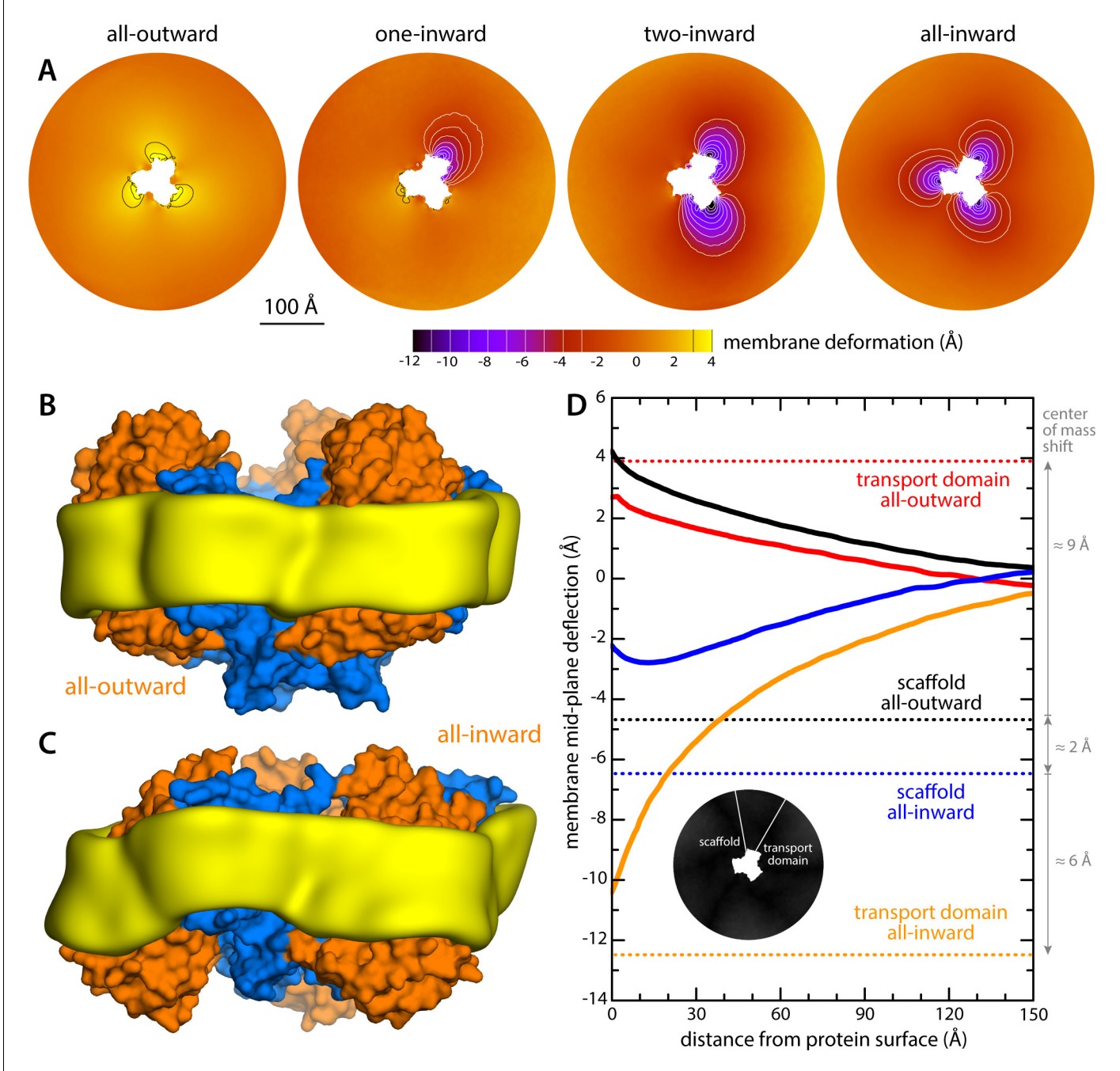

**Figure 2.** Changes in membrane morphology induced by the conformational cycle of Glt$_{Ph}$. The results are based on coarse-grained MD simulations of the transporter in a POPC bilayer at 298K. (**A**) Deflection of the membrane mid-plane for each of the primary states in the cycle. The deflection is quantified by calculating the mean value of the $Z$ coordinate of the bilayer across the $X$-$Y$ plane. The zero-level is set at ~200 Å from the protein center, where the membrane mid-plane is flat, on average. The resulting maps are viewed from the extracellular space. Each map is the mean of $N = 3$ observations, each of which is a time-average for one simulated trajectory. Positive values reflect an outward deflection; negative values reflect inward bending. Values equal to or greater than ±3 Å are contoured (black/white), for clarity. From left to right, the standard error of the data across each map is, on average, 0.8 Å, 0.8 Å, 0.5 Å and 0.6 Å. (**B**) Structure of all-outward Glt$_{Ph}$ (represented as in *Figure 1*), alongside a calculated density map for the lipid bilayer alkyl chains within 10 Å of the protein surface (yellow), based on all simulation data gathered for this state. See also *Figure 2—video 1*. (**C**) Same as (B), for the all-inward state. See also *Figure 2—video 2*. (**D**) Cross-sections of the membrane-deflection data in (A), plotted as a function of the distance to the protein surface. The cross-sections project away from the transport domain, in either the all-outward or all-inward states (solid red and orange lines, respectively), or from the scaffold domain (solid black and blue lines, respectively), following the direction indicated by the inset schematic. Horizontal dashed lines indicate the location of the center of mass of each domain, in either conformation (same color scheme).

*Figure 2 continued on next page*

*Figure 2 continued*

The online version of this article includes the following video(s) for figure 2:

**Figure 2—video 1.** Changes in membrane morphology induced by the conformational cycle of Glt$_{Ph}$.

https://elifesciences.org/articles/50576#fig2video1

**Figure 2—video 2.** Changes in membrane morphology induced by the conformational cycle of Glt$_{Ph}$.

https://elifesciences.org/articles/50576#fig2video2

## Transport domains bend the membrane, while scaffold domains anchor it

Although the coarse-grained MARTINI forcefield yields a reasonably accurate description of membrane elasticity (*Marrink et al., 2004*; *Fiorin et al., 2019*), the degree to which it also captures the specificity of bi-molecular interactions has been questioned (*Javanainen et al., 2017*). We thus considered the possibility that the nature of the membrane deformations observed for the inward-facing conformation of Glt$_{Ph}$ results from the lack of sufficient detail in the representation of protein and lipid structures and their interactions. To address this concern, we carried out three independent simulations of the all-inward state, using the all-atom CHARMM forcefield (see Materials and methods). Each trajectory was initialized with a different starting condition, derived from the previous coarse-grained simulations. The simulation systems are therefore identical in size to those described above. Of course, the computational cost is much greater, as the simulation system now amounts to ~3,000,000 atoms (*Figure 1—figure supplement 1B*).

Reassuringly, the results obtained with the all-atom representation recapitulate those obtained with the coarse-grained forcefield (*Figure 5*). Even after a suitable relaxation time following the change in forcefield (*Figure 5—figure supplement 1AB*), and despite a significant number of exchanges between lipids in the bulk and near the protein surface (*Figure 5—figure supplement 1C*), the deformations induced by the transport domains were sustained in both magnitude and range; the abrupt restoration of membrane shape at the scaffold-domain interface also continued to be observed.

What might explain these striking effects? The complementary analyses described in *Figure 5C* and *Figure 6* indicate that the membrane adapts to the conformational state of the protein to preclude strongly hydrophilic side chains from penetrating the core of the bilayer, while also avoiding exposure of hydrophobic portions of the protein surface to solvent. The deformed state of the bilayer also appears to be stabilized by numerous hydrogen-bonding interactions between polar and aromatic residues and lipid headgroups. For example, it is apparent that Arg105 and Lys230, which face the extracellular solution in the outward-facing state (*Figure 1*), would be fully dehydrated in

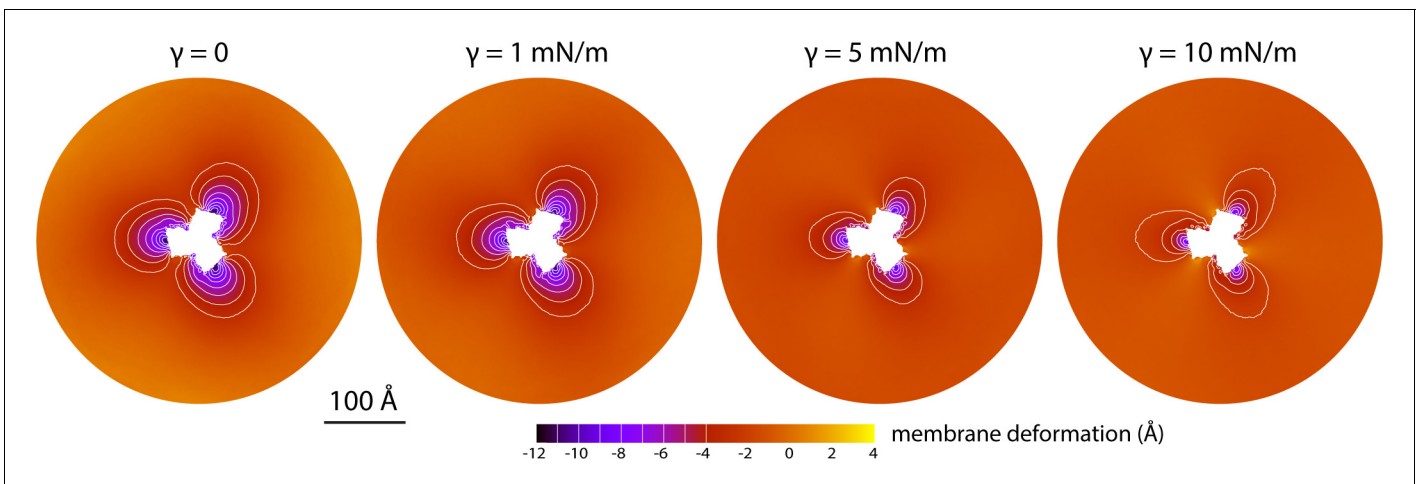

**Figure 3.** Membrane deformation induced by all-inward Glt$_{Ph}$ in coarse-grained MD simulations in a POPC bilayer at 298 K, with and without an applied membrane tension of increasing magnitude (as indicated). The deflection of the membrane mid-plane was calculated and represented as in *Figure 2A*. From left to right, the standard error of the data ($N$ = 3) across each map is, on average, 0.6 Å, 0.7 Å, 0.4 Å and 0.3 Å.

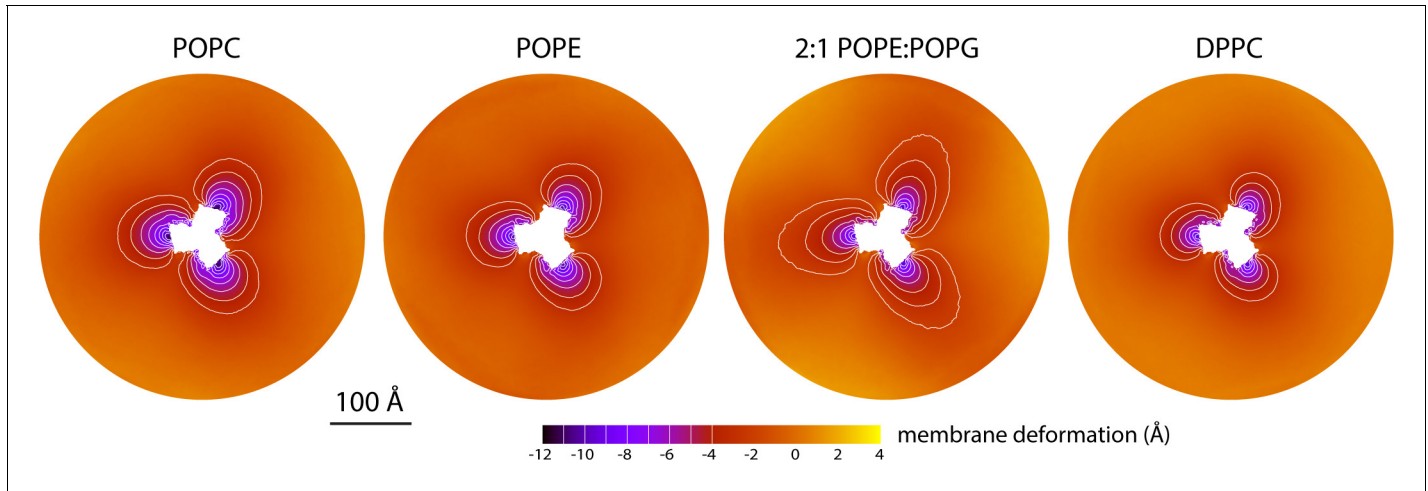

**Figure 4.** Membrane deformation induced by all-inward Glt_Ph in coarse-grained MD simulations in different bilayers. The data for POPC, POPE, and 2:1 POPE:POPG were obtained at 298 K; the data for DPPC were obtained at 323 K. The deflection of the membrane mid-plane was calculated and represented as in *Figure 2A*. From left to right, the standard error of the data (N = 3) across each map is, on average, 0.6 Å, 0.6 Å, 0.4 Å and 1.0 Å. The online version of this article includes the following figure supplement(s) for figure 4:

**Figure supplement 1.** Changes in bilayer thickness induced by the conformational cycle of Glt_Ph.
**Figure supplement 2.** Changes in lipid-chain tilt induced by the conformational cycle of Glt_Ph.

the inward-facing conformation were it not for the fact that the membrane bends inwards (*Figure 6C*). Instead, these residues form highly persistent interactions with the phosphate and ester groups in the lipid bilayer (*Figure 5C*). Conversely, a large number of hydrophobic residues on the intracellular side of the transport domain would be exposed to solvent if the membrane was unchanged (*Figure 6C*). As illustrated in *Figure 6B* (and *Figure 6—figure supplement 1B*) individual per-residue preferences accumulated across the protein surface add up to very large energetic gains (which, as will be discussed later, are balanced by the resistance of the membrane to be deformed). Notably, this pattern of H-bonding interactions extends to the scaffold, where they appear to help anchor the bilayer in between adjacent transport domains, on both sides of the membrane (*Figure 5C*). Interestingly, clear co-evolutionary relationships can be detected across Glt_Ph homologs for several clusters of membrane-exposed residues that include those engaged in hydrogen-bonding with lipids in our simulations (*Figure 6D*). In summary, the changes in membrane morphology induced by the conformational cycle of Glt_Ph thus appear to be dictated by the displacement of protein interfaces that have evolved to be closely matched with the chemical features of the membrane and the surrounding solvent.

## The inward-facing state incurs a major energetic penalty due to membrane bending

The data presented in *Figures 2–5* shows each Glt_Ph protomer causes a membrane deflection of about 10 Å in an area of about 1,000 Å$^2$, that is, a very pronounced change in membrane curvature. This observation begs the question: what is the energetic cost of such a deformation? To answer this question precisely is very challenging. As we discuss in detail elsewhere (*Fiorin et al., 2019*), most computational strategies to examine the energetics of membrane bending are based upon the Helfrich-Canham theory, which predicts a relationship between curvature and energy dictated by the bending modulus of the bilayer (*Canham, 1970*; *Helfrich, 1973*). Although this theory is appropriate for mesoscopic perturbations, it can be inadequate on its own in the length scales that are relevant to membrane protein mechanistic studies (*Goetz et al., 1999*; *Brannigan and Brown, 2006*). Hence, a number of variations and extensions of the Helfrich-Canham function have been proposed to account for other possible contributions to the membrane energetics, based on more complex functions of the local bilayer curvature (*Brannigan and Brown, 2006*; *Watson et al., 2012*; *Khelashvili et al., 2013*). An alternative route to evaluate membrane perturbations would be to calculate the associated free-energy cost directly from a molecular dynamics simulation, using

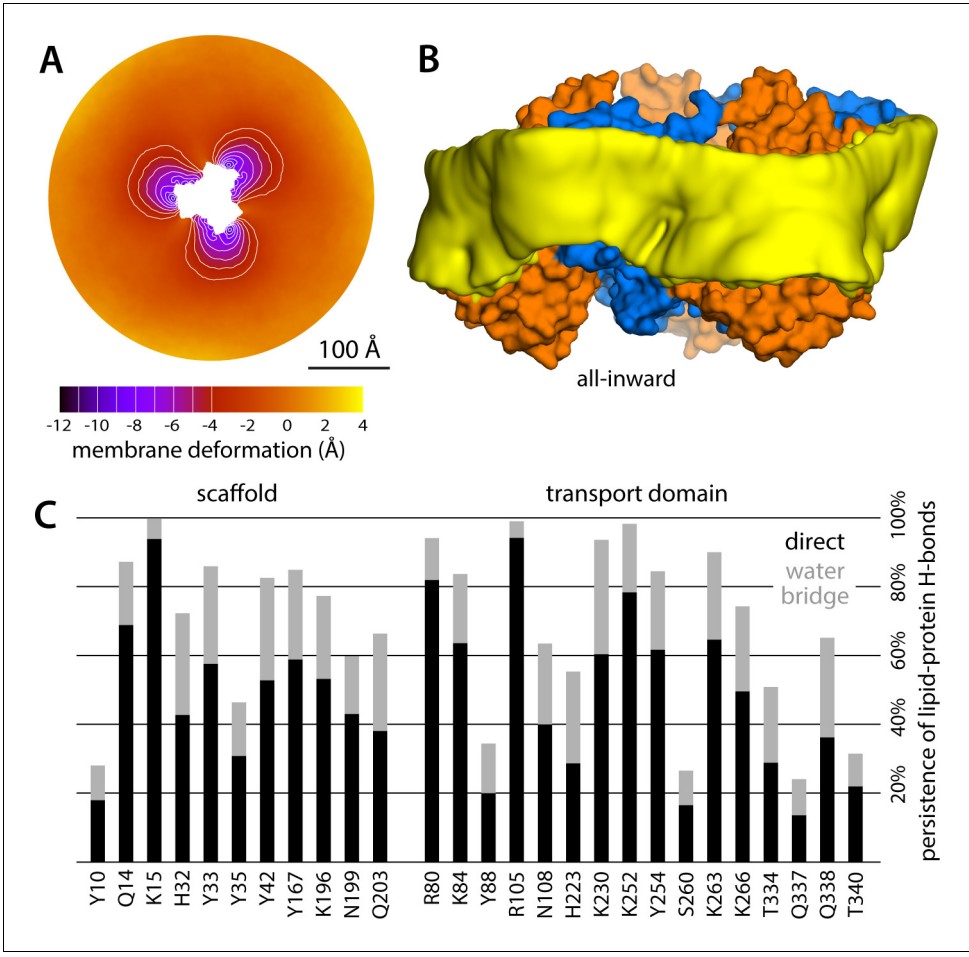

**Figure 5.** Membrane deformation induced by all-inward Glt$_{Ph}$, based on large-scale all-atom simulations in DPPC at 323 K. (**A**) Deflection of the membrane mid-plane relative to a flat surface, calculated exactly as in **Figure 2A**. The standard error of the data ($N$ = 3 trajectories, 150 ns each) across the deflection map is, on average, 1.5 Å. (**B**) Structure of all-inward Glt$_{Ph}$ (as in **Figure 1**), alongside a calculated density map for the lipid bilayer alkyl chains within 10 Å from the protein surface (yellow), based on all simulation data gathered for this state. See also **Figure 5—video 1**. (**C**) Hydrogen-bonding lipid-protein interactions observed during the all-atom simulations of all-inward Glt$_{Ph}$. Direct donor-acceptor interactions are considered, as are interactions mediated by a water molecule. For each protein side chain, the plot quantifies the fraction of the simulation time during which an interaction with a lipid was observed. **Figure 1C** indicates the location of most of the side chains observed to have persistent lipid interactions.

The online version of this article includes the following video and figure supplement(s) for figure 5:

**Figure supplement 1.** Evaluation of potential biases resulting from conversion of coarse-grained molecular configurations into an all-atom representation.

**Figure 5—video 1.** Membrane deformation induced by all-inward Glt$_{Ph}$ in large-scale all-atom simulations.

https://elifesciences.org/articles/50576#fig5video1

enhanced-sampling techniques. This type of model-free, microscopic approach has become state-of-the-art in computational studies of other molecular-scale processes such as ion permeation, ligand binding, or protein conformational change (**Perez et al., 2016**; **Harpole and Delemotte, 2018**; **Flood et al., 2019**), superseding other types of models and theories. The reason why microscopic enhanced-sampling approaches have lagged behind for membranes is the difficulty in formulating appropriate descriptors of the membrane shape and configuration – that is the so-called 'collective variable' problem.

In a recent development, we have reported a novel free-energy simulation strategy to address this problem, which we refer to as Multi-Map (**Fiorin et al., 2019**). The central element of this

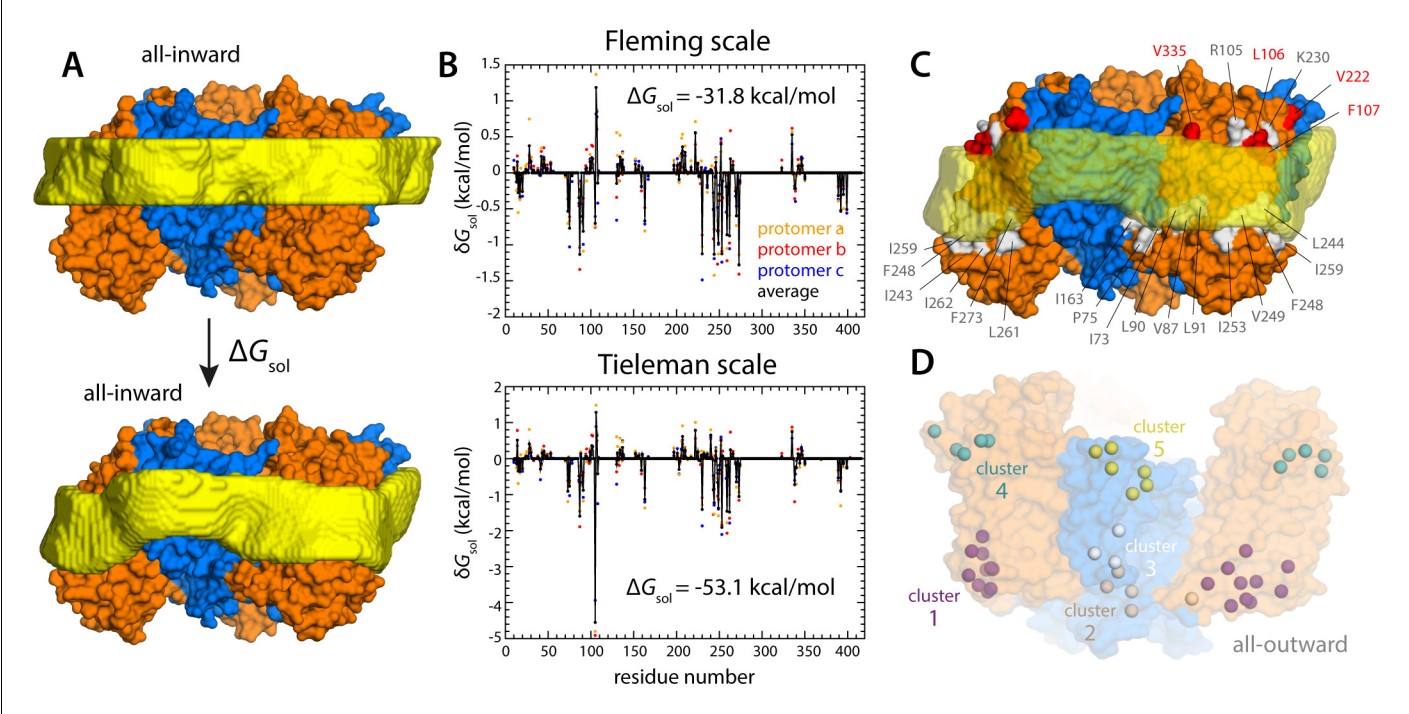

**Figure 6.** Energetics of solvation and evolutionary conservation of the Glt$_{Ph}$ lipid interface. (**A**) Molecular systems used to evaluate the change in the free energy of polar/hydrophobic solvation that results from membrane bending, for all-inward Glt$_{Ph}$. The solvent-accessible surface area of each residue in the protein was calculated for either case (Materials and methods). (**B**) Per-residue change in the free-energy of polar/hydrophobic solvation, deduced from two alternative hydrophobicity scales (Materials and methods). Negative values of $\delta G_{sol}$ indicate the deformed membrane state is favored; positive values favor the flat configuration instead. All residue contributions, in the three protomers, were summed to obtain the total value of $\Delta G_{sol}$. (**C**) Residues for which the magnitude of $\delta G_{sol}$ is 1 $k_B T$ or greater (with both scales) are highlighted in the context of the proposed membrane deformation for all-inward Glt$_{Ph}$. Residues that favor the deformed state are shown in gray; those that favor the flat state are shown in red. The protein structure examined in panels (A, B) is an equilibrated snapshot of the all-atom simulation of all-inward Glt$_{Ph}$. An analogous analysis of the X-ray structure of all-inward Glt$_{Ph}$ is shown in *Figure 6—figure supplement 1*. (**D**) Residues involved in H-bonds to lipid head groups (*Figure 5C*) that are also predicted to have co-evolved with neighboring residues at the protein-lipid interface (Materials and methods). The position of these residues in the outward-facing X-ray structure of Glt$_{Ph}$ is indicated by their Cα atoms (spheres). On the cytoplasmic side of the protein, there are three main clusters: one on the transport domain (cluster 1, purple) comprising residues E80, K84 and Y88 (TM3), L250, Y254 (TM6), I411, V412, K414, T415, and E416 (TM8); and two mostly on the scaffold: one including A67 (TM3), A164, Y167 (TM4), K196, and G200 (TM5) (cluster 2, gray), and the other including K15 (TM1), Q203 and I207 (TM5) (cluster 3, white). On the periplasmic side, there are two clusters: one on the transport domain (cluster 4, cyan), comprising R105, N108 (TM3), F323 (TM7), V335, and Q338 (HP2a); and one on the scaffold (cluster 5, yellow) containing L30, H32, Y33 (TM1), T41, Y42, and V43 (TM2). The online version of this article includes the following figure supplement(s) for figure 6:

**Figure supplement 1.** Energetics of polar and hydrophobic solvation of the Glt$_{Ph}$ lipid interface.

method is a collective variable that quantifies the similarity between the instantaneous configuration of the lipid membrane and a set of pre-defined density distributions mapped onto a 3D grid. Biased exploration of this Multi-Map variable, for example using umbrella-sampling simulations, not only transforms the bilayer shape as dictated by the set of target density distributions, but also permits derivation of the corresponding potential-of-mean-force. What, then, is the free-energy cost of the membrane deformations induced by inward-facing Glt$_{Ph}$?

To evaluate this cost, we devised a Multi-Map calculation whereby a lipid membrane is driven to deform in a manner that mimics the perturbation caused by Glt$_{Ph}$, but in the absence of the protein. In *Figure 7A*, we show 2D representations of three membrane configurations sampled by this enhanced-simulation methodology. The simulations also sample configurations for which the amplitude of the membrane perturbations is greater and smaller than those shown in *Figure 7A*, that is larger and smaller values of the Multi-Map variable. The corresponding potential-of-mean-force curve, that is, the free-energy change as a function of the deformation amplitude, is shown in *Figure 7B*. From comparison of this data with the results shown in *Figure 2A*, it can be inferred that

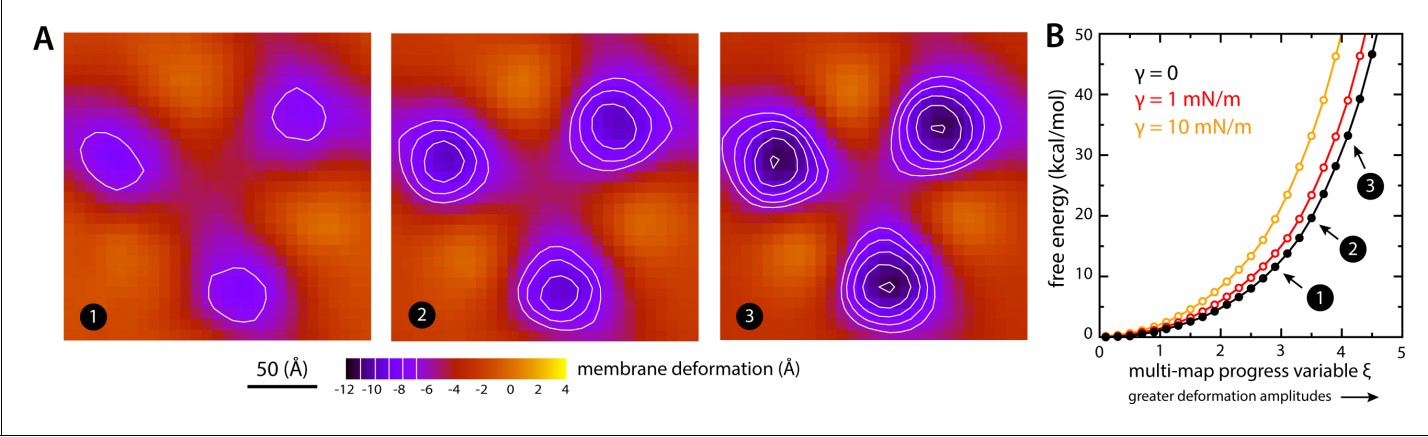

**Figure 7.** Estimate of the free-energy cost associated with the membrane deformation caused by all-inward Glt$_{Ph}$, from direct potential-of-mean-force calculations. (**A**) Simulated membrane deformation, in the absence of the protein, induced by application of the Multi-Map method in combination with umbrella sampling, for a coarse-grained POPC lipid bilayer at 298 K. The figure shows three deflection maps analogous to those shown in *Figure 2A*, that is, calculated from trajectory data by averaging the $Z$ coordinate of the bilayer mid-plane across the range of $X$ and $Y$ encompassed by the simulation box. The deflection maps shown correspond to three individual umbrella-sampling windows used in this free-energy calculation, differing in the amplitude of the perturbation that is induced in each case. Other trajectories/windows sample deformation amplitudes that are smaller or larger than those represented in the figure, that is, smaller or larger values of the Multi-Map variable. Each map reflects an average of 18 independent simulations of 1 μs each. (**B**) Potential-of-mean-force (PMF) curve for the morphological perturbation depicted in panel (A), as a function of the Multi-Map variable, that is, as a function of an increasing deformation amplitude. The free-energy values for the three configurations represented in panel (A) are indicated. PMF curves are also shown for two additional calculations based on umbrella-sampling simulations under an applied membrane tension, for the values indicated. Each of these PMF curves is an average of 18 independent calculations, each sampling 1 μs per window. Error bars for each curve average to about 0.6 kcal/mol.

The online version of this article includes the following figure supplement(s) for figure 7:

**Figure supplement 1.** Comparison of membrane-bending free-energy values calculated with the Multi-Map method and with the Helfrich-Canham theory, for the same ensembles of molecular configurations.

the cost of the membrane perturbation induced by all-inward Glt$_{Ph}$ is on the order of 20 kcal/mol in total, or 6–7 kcal/mol per protomer. (This value was obtained in the absence of membrane tension; under membrane tension, the cost would be greater, as shown in *Figure 7B*, but the extent of the deformation is smaller, as shown in *Figure 3*.) Clearly, this energetic penalty is sizable; as noted above, however, energetic gains that result from polar and hydrophobic solvation of the protein surface when the membrane is deformed are comparable in magnitude, if not greater (*Figure 6*). At any rate, this analysis shows that the morphological preference of the membrane is a major contributor to the free-energy landscape of this transporter; it specifically and strongly opposes the inward-facing state, and so must be counterbalanced by other free-energy contributions for the transporter to carry out its structural mechanism. (For completeness, a comparative analysis with results obtained using the Helfrich-Canham macroscopic theory is presented and discussed in *Figure 7—figure supplement 1*.)

## Discussion

Our simulations predict that the conformational cycle of Glt$_{Ph}$ induces a long-ranged remodeling of the surrounding lipid membrane, as a result of the elevator-like motion of the transport domains, each of which displaces its protein-lipid interface by about one-half the width of the bilayer hydrophobic core. Strikingly, this perturbation is abruptly suppressed at the point where the bilayer meets the trimerization domain; therefore, each protomer induces an independent deformation, consistent with existing experimental evidence demonstrating no protomer-protomer cooperativity in this class of transporters (*Grewer et al., 2005*; *Koch and Larsson, 2005*; *Koch et al., 2007*; *Leary et al., 2007*; *Akyuz et al., 2013*; *Erkens et al., 2013*; *Akyuz et al., 2015*; *Ruan et al., 2017*). These results provide a working hypothesis and as such require experimental verification; while direct structural

data confirming the drastic effects that we predict here are still lacking, such information is not beyond reach. As demonstrated by recent structural studies of $Ca^{2+}$-dependent lipid scramblases, single-particle cryo-EM can reveal the morphological features of detergent micelles and lipid nano-discs in considerable detail (*Falzone et al., 2018*; *Falzone et al., 2019*; *Kalienkova et al., 2019*); crystallographic studies can also reveal the contours of the lipid bilayer, as observed for stacked membrane crystals of the SERCA $Ca^{2+}$-ATPase pump (*Sonntag et al., 2011*). Analogous data for alternate conformational states of $Glt_{Ph}$, or a close homolog thereof, ought to validate or refute the predicted impact of this transporter on the membrane morphology.

Single-molecule FRET measurements of substrate-loaded and substrate-free $Glt_{Ph}$ in outside-out proteoliposomes have indicated that the intrinsic free-energy difference between the outward- and inward-facing states of the transporter is approximately zero, that is, both states are approximately equally populated (*Akyuz et al., 2013*; *Erkens et al., 2013*; *Akyuz et al., 2015*). Here, we have shown that the membrane deflection induced by inward-facing $Glt_{Ph}$ translates into a substantial energetic penalty, which we estimate to be in the order of 6–7 kcal/mol per protomer, or about 20 kcal/mol in total. To balance out this cost, therefore, the inward-facing conformation must somehow recoup this energy, through distinct protein-protein, protein-lipid and/or protein-water interactions. Admittedly, at this point we can only infer the magnitude of free-energy penalty from simulations where the Multi-Map method mimics the impact of the protein on the membrane. While imperfect, it is worth underscoring the technical breakthrough made by this approach: it provides a means to estimate the membrane energetics directly from simulated molecular-dynamics trajectories, without a priori theoretical assumptions. Thus, we believe that further methodological developments and systematic applications of this methodology, in combination with other enhanced-sampling techniques, will result in increasingly precise estimates, and will also enable us to dissect the compensating interactions that must occur in this and other membrane-protein systems. These caveats notwithstanding, we believe our estimate of the free-energy penalty of membrane bending not to be at all unrealistic. For example, based on electrophysiological studies of the gating mechanism of the *Shaker* voltage-gated $K^+$ channel, it has been deduced that at zero membrane potential the conformational free-energy of the open state is about 15 kcal/mol lower than that of the closed state (*Chowdhury and Chanda, 2012*). Similarly, a value of 16 kcal/mol was deduced for the $Na^+$-channel $Na_v1.4$ (*Chowdhury and Chanda, 2012*). Regardless of the specifics, this study underscores that the configurational energetics of the lipid bilayer are non-negligible and must therefore be incorporated into the conceptual models and theories used to describe membrane-protein mechanisms – much in the same way one would consider, for example, the dehydration energetics of different ions (also in the order of tens of kcal/mol) when rationalizing the selectivity or conductance rates of a channel protein. It is hoped, therefore, that the calculations presented here will spur further biophysical studies designed to assess the impact of bilayer composition and morphology on the energetics of transporters – in analogy with research other membrane proteins such as receptors (*Brown and Chawla, 2017*) and channels (*Phillips et al., 2009*; *Andersen, 2013*). For example, it would be of interest to dissect how the functional dynamics of a transporter such as $Glt_{Ph}$ are influenced by the amino-acid make-up of the protein-lipid interface, as well as by lipid bilayer composition. Indeed, functional studies have shown that subtle lipid modifications (methylation) and single-point mutations (Tyr33) have detectable effects on the rates of $Glt_{Ph}$ transport (*McIlwain et al., 2015*). It is also intriguing that long-chain polyunsaturated fatty acids (PUFA) modulate the activity of neuronal EAAT transporters (*Zerangue et al., 1995*; *Fairman et al., 1998*; *Tzingounis et al., 1998*; *Grintal et al., 2009*) and are also known to influence the bending energetics of lipid bilayers (*Cordero-Morales and Vásquez, 2018*).

The aforementioned smFRET studies have also indicated that the balance of outward- and inward-facing states hardly differs when measured in liposomes (containing mostly PE, PG and PC lipids) or in DDM micelles (*Akyuz et al., 2015*). Taken together with our findings, this observation challenges the notion that detergent micelles have no distinct morphological preference and will comply to the conformation of a protein at little or no energetic cost. To the contrary, when a detergent solution is concentrated above a certain threshold, the micelles that form have an inherently preferred geometry and size (*Lipfert et al., 2007*; *Oliver et al., 2013*), that is, there exists a well-defined free-energy minimum of micelle formation. When a micelle assembles around a protein, one might expect that free-energy minimum to naturally shift, that is, the micelle morphology will adapt to the volume and surface of the protein. However, this adaption is not necessarily identical for all

conformations of a protein; indeed, one might expect that different free-energy minima will exist for alternative structural states (both in magnitude and morphology), and that this difference in will be greater for some detergents than for others. The expectation that the deformation of a micelle entails little or no energetic cost is even less intuitive if one assumes that the micelle does not reassemble when a protein changes structure. Indeed, the above-mentioned cryo-EM data for nhTMEM16 shows that a highly localized feature of the protein surface causes nearly identical morphological changes in a lipid nanodisc and in a DDM micelle, namely a perturbation that gradually decays along the protein perimeter (*Kalienkova et al., 2019*). If the DDM micelle was indeed much softer or more compliant than the lipid nanodisc, the resulting curvature would be more localized. Thus, on this matter too we believe our data highlights the need for further work probing how the thermodynamics of micellar or lipid solvation might depend on the conformational state of a protein, or on its oligomerization state.

The dynamics of Glt$_{Ph}$ at the single-molecule level have also been evaluated using high-speed AFM measurements in protein-dense 2D preparations (*Ruan et al., 2017*). These elegant measurements clearly confirmed that each protomer exchanges between outward- and inward-facing states stochastically (provided the adequate occupancy state) and independently from each other. Intriguingly, though, in these measurements the population ratio between outward- and inward-facing is shifted significantly against the latter, by about 5-fold (*Ruan et al., 2017*; *Heath and Scheuring, 2019*). A possible explanation for the discrepancy between the smFRET and AFM measurements is that the curvature of the outside-out proteoliposomes used in the former experiment favors the inward-facing state, while the flat membranes in the latter do not. However, our finding that the membrane perturbation caused by Glt$_{Ph}$ projects away from the protein surface for several nanometers provides an alternative explanation: transition to the inward-facing state would be more energetically costly when this deformation must occur in a more confined space. In other words, it is conceivable that the aforementioned balance between large, competing energetic contributions is altered as a result of molecular crowding, in this case favoring the outward-facing conformation. Interestingly, some members of the EAAT family are expressed in the membranes of certain cell types at very high densities (in the order of thousands of transporters per square-micron [*Danbolt, 2001*]), raising the possibility that crowding effects occur in physiological settings. On a related, more technical note, it is worth noting that molecular simulation studies aiming to evaluate the alternating-access mechanism and protein-lipid interplay for cases such as Glt$_{Ph}$ might be significantly skewed by finite-size effects if the lipid bilayer patch is not sufficiently large to accommodate the full range of the membrane perturbation created by the protein.

Large-scale structural changes are not exclusive to membrane proteins in the EAAT superfamily. Elevator-like secondary-active transporters are, however, appealing model systems to evaluate the interplay between proteins and the lipid bilayer, given the range of their motions and that these motions occur spontaneously and, in some cases, on time-scales amenable to single-molecule measurements like smFRET or high-speed AFM. As an example of another elevator-like mechanism, an initial analysis of the bacterial sodium-dicarboxylate symporter VcINDY is shown in *Figure 8*. VcINDY is a dimer, but is similar to Glt$_{Ph}$ in that each of the constituent protomers features two distinct domains, one facilitating dimerization and the other responsible for substrate translocation (*Mancusso et al., 2012*; *Mulligan et al., 2016*). Our simulations show that upon transition from the (predicted) outward-facing state to the (experimentally determined) inward-facing conformation, the transport domains would induce a deformation in the membrane that is comparable to that observed for Glt$_{Ph}$. Again, this deformation is suppressed by the scaffold, suggesting that the protomers function independently. Despite the commonalities between Glt$_{Ph}$ and VcINDY, however, it is important to note that other elevator-like transporters, perhaps even those in the same structural family, might influence the membrane differently, as this interplay is ultimately determined by the amino-acid make-up of the protein surface. For example, it is entirely possible that, in some cases, the outward-facing state perturbs the membrane the most. To document and dissect this variability is no doubt of interest and will be the focus of future studies.

Glt$_{Ph}$ and VcINDY are stark examples of membrane proteins that cause large-scale morphological changes in the bilayer. Needless to say, numerous channels and transporters also undergo major conformational changes during function, which might impact the membrane to varying degrees. The computational analysis presented here makes it clear that the inherent configurational energetics of

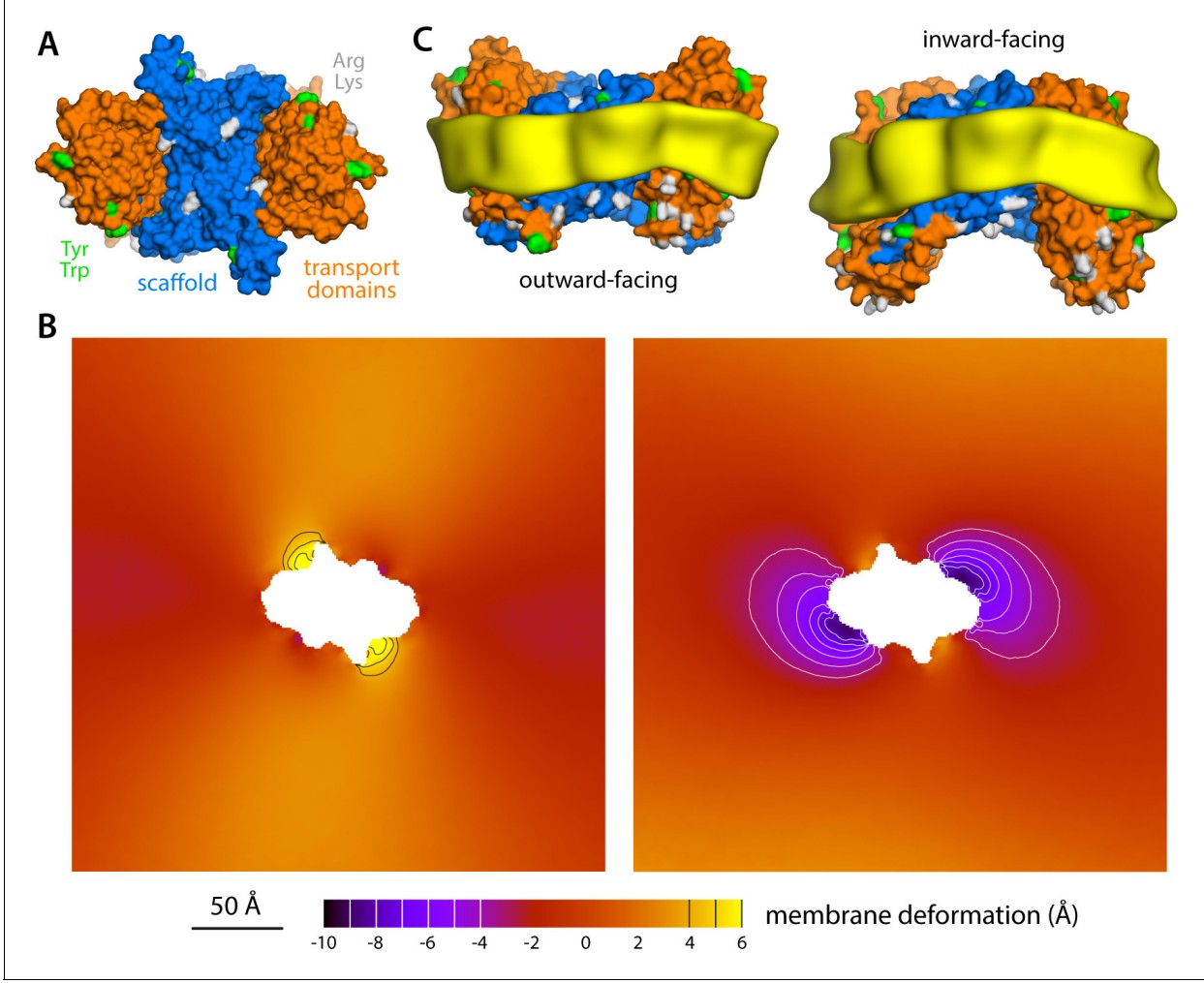

**Figure 8.** Membrane deformation induced by the Na+-dicarboxylate symporter VcINDY, based on coarse-grained MD simulations in POPC at 298 K. (**A**) Structure of the VcINDY dimer in the outward-facing state, viewed along the membrane perpendicular from the extracellular space. The protein is represented as Glt$_{Ph}$ in *Figure 1*. (**B**) Deflection of the membrane mid-plane induced by VcINDY in the outward- and inward-facing states (left and right, respectively) based on three independent simulations for either state. The view is from the extracellular space. From left to right, the standard error of the data ($N = 3$) across each map is, on average, 1.0 Å and 0.8 Å. (**C**) Structures of outward- and inward-facing VcINDY (represented as in panel A), alongside calculated density maps for lipid alkyl chains within 10 Å from the protein surface (yellow), based on all the simulation data obtained for either state. Note the outward-facing state is a model, constructed on the basis of the experimental inward-facing structure through repeat-swapping (*Mulligan et al., 2016*).

the lipid bilayer can be both non-negligible and markedly state-dependent, and must therefore be integrated in our conceptualizations of membrane protein mechanisms.

## Materials and methods

### Coarse-grained simulations of Glt$_{Ph}$ trimers and VcINDY dimers

Four different conformational states of the Glt$_{Ph}$ transporter were simulated using a coarse-grained (CG) representation of the protein and its environment, using the MARTINI 2.1 forcefield (*Marrink et al., 2007*). Symmetric trimers of outward- and inward-facing conformers were obtained from X-ray crystal structures (PDB 2NWL and 3KBC, respectively *Boudker et al., 2007*; *Reyes et al., 2009*). These differ primarily in the position of the transport domains (defined here as residues 74–129 and 223–416), relative to the central oligomerization domain, or scaffold (residues 6–73 and 120–222). Two asymmetric trimers were also considered, combining two outward- and one inward-

facing protomer, or two inward- and one outward-facing protomer. These intermediates were constructed by superposing the scaffold domain of individual protomers from the inward-facing structure onto the trimer with all protomers outward-facing. All four states of the transporter were embedded in a hydrated bilayer of 1-palmitoyl-2-oleoyl-sn-glycero-3-phosphocholine (POPC) lipids, to evaluate the morphological adaption of the membrane to the protein conformation. The conformation of each state was maintained throughout the simulations by applying a network of harmonic distance restraints to a pre-defined set of pairs of non-bonded protein atoms. This set comprises all pairs of backbone atoms separated by a distance between 5 and 9 Å; the force constant of these elastic restraints is 1.2 kcal/mol Å$^{-2}$. The protein-membrane systems include ~7500 lipids and ~170,000 solvent molecules, for a total of ~260,000 particles. The dimensions of the simulation systems are approximately $500 \times 500 \times 120$ Å. Counter-ions were added to the solvent to neutralize the total charge of all molecular systems.

All CG simulations were carried out using GROMACS 4.5.5 (*Hess et al., 2008*), with a 10-fs integration time-step. Temperature and pressure were maintained constant using the Berendsen barostat and thermostat (*Berendsen et al., 1984*). The pressure (one atm) was applied semi-isotropically, that is $X$ and $Y$ dimensions (the bilayer plane) fluctuate but at constant $X/Y$ ratio, while fluctuations in $Z$ are independent. Unless specified otherwise, the pressure components $P_{xx}$, $P_{yy}$ and $P_{zz}$ were such that the resulting membrane tension is zero. Periodic boundary conditions were used. Non-bonded interactions were described by a shifted Lennard-Jones potential, cut-off at 12 Å, and by a Coulombic potential with relative dielectric constant $\varepsilon_r$ = 15.

For each Glt$_{Ph}$ state, we calculated three independent trajectories of 600 ns. Analogous, triplicated simulations of 600 ns were also carried out for all-inward Glt$_{Ph}$ in a bilayer of 1,2-dipalmitoyl-sn-glycero-3-phosphocholine (DPPC) lipids at 323 K; in a mixed bilayer of 1-palmitoyl-2-oleoyl-sn-glycero-3-phosphoethanolamine (POPE) lipids and 1-palmitoyl-2-oleoyl-sn-glycero-3-phospho-(1'-rac-glycerol) (POPG) lipids in a 2:1 ratio, at 298K; and in a bilayer of POPE lipids, also at 298 K. Triplicated simulations of 600 ns were also carried out for all-inward Glt$_{Ph}$ in POPC at 298 K under applied membrane tensions of 1, 5 and 10 mN/m. Finally, triplicated simulations of 600 ns each were carried out for the Na$^+$-dicarboxylate transporter VcINDY, in POPC at 298 K, in both outward- and inward-facing conformations; the latter corresponds to the experimentally determined X-ray structure (PDB 5ULD, *Nie et al., 2017* ) while the former is a computational model (*Mulligan et al., 2016*).

## All-atom simulations in all-inward Glt$_{Ph}$

For the all-inward state (with all three Glt$_{Ph}$ protomers inward-facing) we also calculated three independent trajectories of 150 ns each, using an all-atom representation of the molecular system. The starting coordinates for each of these simulations were obtained from selected snapshots of the three independent CG simulations carried out for the all-inward state in DPPC. Specifically, we chose those snapshots in which the instantaneous shape of the membrane mid-plane showed the lowest RMS deviation from the shape calculated by averaging all trajectory data for this all-inward state. Using an all-atom model of the protein derived from the crystal structure (PDB 3KBC), the surrounding (coarse-grained) membrane and solvent were transformed into an all-atom representation as prescribed elsewhere (*Wassenaar et al., 2014*), and equilibrated through a series of simulations implementing positional restraints. The final models comprise ~3,000,000 atoms. The all-atom simulations were carried out using NAMD version 2.9 (*Phillips et al., 2005*). The CHARMM36 force field (*Klauda et al., 2010*; *Best et al., 2012*) was used for the protein and the lipids, while the TIP3P model was adopted for the water (*Jorgensen et al., 1983*). To preserve the conformation of the protein, a restraint was applied to the RMSD of the C$_\alpha$ trace, relative to the experimental structure, of force constant 100 kcal/mol Å$^{-2}$. The simulations were carried out at constant temperature (323 K), using a Langevin thermostat, and constant semi-isotropic pressure (one atm), using the Nosé-Hoover-Langevin barostat, with periodic boundaries. The integration time-step was 2 fs. Electrostatic interactions were calculated using Particle-Mesh Ewald with a real space cut-off of 12 Å. The same cut-off was also used for the shifted Lennard-Jones potential describing van der Waals interactions.

## Energetics of polar/hydrophobic solvation

The molecular systems depicted in *Figure 6A* show two membrane states: one deformed, and identical to that observed in our simulations (*Figure 5A*), and a hypothetical version thereof with the

same thickness (16 Å) but flat; both membranes are aligned at the scaffold. (In this context, 'membrane' refers to the solvent-excluded region of the bilayer). For each of these two systems, we calculated the solvent-accessible surface area (SASA) for each of the protein residues $i$, denoted by $A_{bent}(i)$ and $A_{flat}(i)$, using PyMol (Schrödinger, Inc) and a probe radius of 1.4 Å. A SASA value denoted as $A_{max}(i)$ was also calculated for each amino-acid type when fully solvent-exposed in a GXG tri-peptide. The per-residue change in the free-energy of polar/hydrophobic solvation (*Figure 6B*) was then calculated as $\delta G_{sol}(i) = S_{transfer}(i) \times [ A_{flat}(i) - A_{bent}(i) ] / A_{max}(i)$, where $S_{transfer}(i)$ are identical for residues of the same type, and derive from two independent hydrophobicity scales: one based on experimental measurements of the stability of wild-type and mutagenized OMPLA, referred to as the Fleming scale (*Moon and Fleming, 2011*); and another based on PMF calculations of side-chain analog insertion in a DOPC bilayer, referred to as the Tieleman scale (*MacCallum et al., 2007*). For His, Gly and Pro, which lack values in the latter scale, values from the Fleming scale were used in both calculations.

## Coevolutionary analysis

The Uniprot sequence GLT_PYRHO was used as input to the EVcouplings server (*Marks et al., 2011*) with default parameters. The results can be considered to be robust because the ratio of effective sequences in the alignment (9565 sequences obtained using a bit-score threshold of 0.3) to the length of the protein sequence is >20. To limit our search to reliable scores, we focused on pairs whose scores have coupling probabilities greater than 0.5, which comprises the top 0.8% of all possible pairs. To identify positions that co-evolved in order to maintain interactions with the lipid head groups, we focused on pairs involving at least one residue known to H-bond with lipids in the all-atom simulations of Glt$_{Ph}$ (*Figure 5C*). Pairs were excluded if they contained buried residues, or residues exposed only to the central aqueous vestibule, or were less than eight residues apart in sequence, that is within two turns of a helix. The remaining pairs were assigned to a cluster if at least one residue was involved in more than one pair in that cluster.

## Potential-of-mean-force (PMF) calculations

The recently developed Multi-Map method (*Fiorin et al., 2019*) was used to compute the free-energy cost of a membrane deformation that is similar to that induced by all-inward Glt$_{Ph}$, but in the absence of the protein. To this end, a set of 10 three-dimensional density maps that idealize this deformation but vary in its amplitude were generated and used to define the so-called Multi-Map coordinate ξ. This variable quantifies the similarity between any instantaneous configuration of the membrane and each of the density maps in the target set. Biased-sampling of the Multi-Map coordinate ξ thus perturbs the membrane as dictated by the set of target maps and permits a derivation of the corresponding free-energy cost as a function of the deformation amplitude, that is, the potential of mean force (*Fiorin et al., 2019*). This technique was applied to a CG bilayer of 1,800 POPC molecules (side length of approx. 230 Å), using a developmental version of NAMD (*Phillips et al., 2005*) and analogous pressure and temperature conditions as those specified above. Umbrella-sampling was employed as the biasing method; the target range in ξ was divided into 61 windows, and 18 simulations of 1 µs each were carried out to sample ξ in each window. The PMF was calculated on the basis of the resulting time-series of ξ, using the WHAM method (*Kumar et al., 1992*). To examine the effect of membrane tension, analogous umbrella-sampling simulations and PMF curves were calculated under applied tensions of 1 and 10 mN/m.

## Acknowledgements

The authors are grateful to numerous colleagues and lab members for their commentary on different aspects of this work – too numerous to be listed here. Special thanks are owed to Janice L Robertson for useful discussions, and to Michael Grabe and co-workers for helping us to apply their continuum-mechanics method to examine the membrane deformations caused by elevator-like transporters. This research was initially supported by the Max Planck Society and The German Research Foundation, and subsequently by the Divisions of Intramural Research of the National Institute of Neurological Disorders and Stroke and of the National Heart, Lung and Blood Institute, National Institutes of Health (NIH). In part, this work utilized the computational resources of the NIH HPC facility Biowulf.

# Additional information

## Competing interests

José D Faraldo-Gómez: Senior editor, *eLife*. Lucy R Forrest: Reviewing editor, *eLife*. The other authors declare that no competing interests exist.

## Funding

| Funder | Author |
| --- | --- |
| National Heart, Lung, and Blood Institute | Wenchang Zhou<br>Giacomo Fiorin<br>Claudio Anselmi<br>José D Faraldo-Gómez |
| National Institute of Neurological Disorders and Stroke | Hossein Ali Karimi-Varzaneh<br>Horacio Poblete<br>Lucy R Forrest |

The funders had no role in study design, data collection and interpretation, or the decision to submit the work for publication.

## Author contributions

Wenchang Zhou, Giacomo Fiorin, Claudio Anselmi, Hossein Ali Karimi-Varzaneh, Horacio Poblete, Formal analysis, Investigation, Visualization; Lucy R Forrest, José D Faraldo-Gómez, Conceptualization, Supervision, Investigation, Visualization

## Author ORCIDs

Wenchang Zhou (iD) http://orcid.org/0000-0003-0397-1032
Giacomo Fiorin (iD) https://orcid.org/0000-0002-8793-8645
Claudio Anselmi (iD) https://orcid.org/0000-0002-3017-5085
Lucy R Forrest (iD) https://orcid.org/0000-0003-1855-7985
José D Faraldo-Gómez (iD) https://orcid.org/0000-0001-7224-7676

## Decision letter and Author response

Decision letter https://doi.org/10.7554/eLife.50576.sa1
Author response https://doi.org/10.7554/eLife.50576.sa2

# Additional files

## Supplementary files

• Transparent reporting form

## Data availability

Input and output files for 1 (out of 3) replica of each simulation system/condition in our study have been uploaded to Zenodo, a public repository free of charge, and is available at the DOI: https://doi.org/10.5281/zenodo.3558957.

The following dataset was generated:

| Author(s) | Year | Dataset title | Dataset URL | Database and Identifier |
| --- | --- | --- | --- | --- |
| Zhou Wenchang, Fiorin Giacomo, Anselmi Claudio, Karimi-Varzaneh Hossein Ali,  Poblete Horacio, Forrest Lucy Rachel, Faraldo-Gómez José Diego | 2019 | Simulation files for "Large-scale state-dependent membrane remodeling by a transporter protein" | http://doi.org/10.5281/zenodo.3558957 | Zenodo, 10.5281/zenodo.3558957 |

The following previously published datasets were used:

| Author(s) | Year | Dataset title | Dataset URL | Database and Identifier |
|---|---|---|---|---|
| Gouaux E, Boudker O, Ryan R, Yernool D, Shimamoto K | 2007 | Crystal structure of GltPh in complex with L-aspartate and sodium ions | https://www.rcsb.org/structure/2NWX | Protein Data Bank, 2NWX |
| Reyes N, Ginter C, Boudker O | 2009 | Crystal structure of GltPh K55C-A364C mutant crosslinked with divalent mercury | https://www.rcsb.org/structure/3KBC | Protein Data Bank, 3KBC |
| Nie R, Stark S, Symersky J, Kaplan RS, Lu M | 2017 | Structure and function of the divalent anion/Na+ symporter from Vibrio cholerae and a humanized variant | https://www.rcsb.org/structure/5ULD | Protein Data Bank, 5ULD |
| Mulligan C, Fenollar-Ferrer C, Fitzgerald GA, Vergara-Jaque A, Kaufmann D, Li Y, Forrest LR, Mindell JA | 2016 | Model of outward-facing VcINDY | http://srv00.recas.ba.infn.it/PMDB/zip_model.php?target=5554 | Protein Model Data Base, PM0080216 |

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
