## [Decision Letter]

**Acceptance summary:**

Conventional molecular dynamics simulations and a new enhanced sampling method, presented in another paper (in *J Comp Chem*), were used to examine the extent of lipid perturbation due to conformational changes of the Na+-aspartate symporter GltPh. This transporter, which follows the 'elevator mechanism', undergoes particularly large conformational changes upon substrate transport; the transition between the outward-facing and inward-facing conformations involves a 15Ang motion of the transport domain with respect to the stationary domain of the protein. Their simulations suggested that this transition radically perturbs the lipid bilayer, which is expected. However, very surprisingly, it leads to a large free energy penalty of about 20 kcal/mol, which of course needs to be balanced by internal components of the free energy. The manuscript addresses a fundamental and very interesting question in membrane biophysics.

**Decision letter after peer review:**

Thank you for submitting your article "Large-scale state-dependent membrane remodeling by a transporter protein" for consideration by *eLife*. Your article has been reviewed by three peer reviewers, including Nir Ben-Tal as the Reviewing Editor and Reviewer #1, and the evaluation has been overseen by Richard Aldrich as the Senior Editor. The following individual involved in review of your submission has agreed to reveal their identity: Simon Scheuring (Reviewer #2).

The reviewers have discussed the reviews with one another and the Reviewing Editor has drafted this decision to help you prepare a revised submission.

Summary:

A new enhanced sampling simulation method, presented in another paper (in review with J Comp Chem), is used to examine the extent of lipid perturbation due to conformational changes of the Na+-aspartate symporter GltPh. This transporter, that typifies the 'elevator mechanism', undergoes particularly large conformational changes upon substrate transport; the transition between the outward-facing and inward-facing conformations involves a 15 Ang motion of the transport domain with respect to the stationary domain of the protein. The simulations suggest that this transition radically perturbs the lipid bilayer, which is expected. However, very surprisingly, it leads to a large free energy penalty of about 20 kcal/mol (which of course needs to be balanced by internal components of the free energy).

Opinion:

The manuscript addresses a fundamental and very interesting question in membrane biophysics. In this respect it is very suitable for publication in *eLife*. The problem is that it is unclear how trustworthy this large estimate is. The following suggestions may increase the credibility of the manuscript.

Essential revisions:

1) The in-plane distributions of the membrane deformations predicted by the simulations are inaccessible for experimental verifications. Moreover, the character of these deformations (types of strains in the case of 3D description or bending/stretching/chain tilting for a 2D description of the membrane) is not determined.

2) "In a recent breakthrough, we have developed and validated a promising free-energy simulation strategy to address this problem, which we refer to as Multi-Map (Fiorin, 2019)". The Fiorin paper is not in review here so we will not comment about it in detail. However, the draft that is included does not consolidate this bold statement. To our understanding it shows that the new formalism converges to the Helfrich-Canham theory and agrees with measurements at long distances (100Ang and more), as it should. However, we did not see any comparison regarding short distances which are relevant here. This statement should be removed.

3) The authors dismiss the Helfrich-Canham theory and its extensions, arguing that they do not hold for short distances. And yet, it would be insightful for the reader to know what values this approximate model and its extensions give for GltPh. At the very least it would indicate how high the 20 kcal/mol value is compared to existing theory.

4) The derived energies of the membrane deformations, which could be indeed important for understanding the effects of the membrane lipid composition on the protein function, are not supported by any independent estimations and remain, therefore, completely dependent on the parameters of the computational model (forcefield parameters), and other details of the simulations. The authors should admit to it or provide experimental measurements.

5) No analysis is provided either of the effects of membrane tension, which may dramatically change the protein-mediated deformations and their distribution, or of the presence in the membrane of specific lipids such as cholesterol, PIPs, DAGs, etc, whose redistribution to the protein vicinity may moderate the elastic energy. This decreases the general impact of the results.

6) Evidently, the authors are aware of the exceedingly high energy penalty due to lipid perturbation and try to explain it in Discussion. However, the arguments raised are not compelling. Thus, the tune of the manuscript should reflect the fact that all we have here is a computation that may or may not be correct.

7) Introduction, third paragraph: The question is nicely set between (1) "moving polar sidechains on their surface into the bilayer interior and exposing hydrophobic ones to the solvent" and (2) "the morphology of the lipid bilayer could adapt to the conformational state of the protein, and match the amino-acid make-up of the protein surface". The authors obviously come to the conclusion that the latter (2) is the case. However, at a considerable (20kcal/mol) cost. Given that the structure of all conformations are known, the authors should provide an estimate of the energetic cost for the rejected hypothesis (1).

The simulations:

8) The boundary conditions imposed on the membrane fragment are not specified. At the same time, in the absence of membrane tension, which seems to be the case here, the effects of the boundary conditions might be substantial.

9) The method is based on combining coarse-grained and all-atom simulations, which can be tricky. In particular, for the all-atom simulations, the authors take the course-grained system they simulated and convert it to all-atom, while also (naturally) changing force-fields. Then they let the system relax and report that the trends observed are consistent with the course-grain simulations. It is theoretically possible that transitioning the system from the course-grained to the all-atom parameters is not that trivial, and maybe the fact that they start from an already equilibrated state introduces some bias. Could it be examined somehow?

10) It is unfortunate that DPPC (transition temperature 41C) was chosen for many simulations. Why? Also, simulations using POPC (transition temperature -2C) are shown. The fact that the results are quasi identical seems rather worrying than comforting. Why these choices? Why not using a mixture, it is not unthinkable that in a mixture the protein would recruit specific lipids on these interfaces between transporter and scaffold domains. Please comment. Or, better, examine in simulations.

Presentation:

11) Introduction: "These perturbations develop to accommodate the amino-acid composition and specific structural features of the protein surface.": This deserves a reference to the Piezo channel, the only protein that displays clear structural features that force the membrane into bending, and for which a theory and experiments about how membrane bending (and flattening / the physics of the membrane) is exploited to gate the channel was presented.

12) The general reader might not know what the second-rank order parameter of the lipid alkyl chains is.

13) Figure 2, Figure 3 and associated, in the context of "Transport domains bend the membrane, while scaffold domains anchor it": While we understand the measurements, the results and the importance of all these results, the presentation is unclear with respect to the setting of the 0-level. From the captions "2(A) The deflection is quantified by calculating the mean value of the Z coordinate of the bilayer across the X-Y plane." and "(A) Deflection of the membrane mid-plane relative to a flat surface". For example, the image in 3A), all the membrane has negative deformation. Wouldn't that correspond to creation of a net elastic and potential energy? Shouldn't the bilayer as a whole still go towards flat and as a result one has negative deformation next to the transporter domains and positive deformation next to the scaffold domain. This does not change anything to the results in terms of local deformation, right? Is the bilayer held in place at the periphery of the simulation box?

14) Further on this: Why is the scaffold domain considered the 'anchor'? Why does the membrane next to the transporter domains (that are in the inward (down) orientation) move down, rather than the scaffold domain moving upwards? Overall – after sufficiently long relaxation of the all-inward structure – shouldn't the overall level of the membrane remain 0 and the membrane next to the transporter domains shift downwards, but the membrane next to the scaffold domains shift upwards? Especially, in light of Figure 3C, which seems to show that there are many more key amino acids in the transporter than in the scaffold domain, one would expect that the transport domain dominates the relative motion. It is unclear how in panels like Figure 2A left, 2A right, Figure 3A, all membrane deformation values can be positive or negative? Is this the reason why such huge energy penalties result from the analysis?

15) Discussion section: The authors find a large energetic penalty for the inward facing state due to membrane deformation. They note that the smFRET studies displayed almost zero energy difference between the states. In contrast the HS-AFM study revealed that the inward facing state indeed was the high energy state (like here). The authors mention that the HS-AFM study was performed in rather densely packed membrane where no such long range lipid relaxations are possible as in the study here, yet it is a flat membrane. In this context, it is also noticeable that the smFRET studies are performed either in detergent or on transporters in small vesicles that are tethered in outside-out configuration only – which goes against the bowl-shaped structure of GltPh – of which the authors see the preference for membrane bending in Figure 1 (side view), which might favor the adoption of the inward-facing state in the smFRET experiments.

---

## [Author Response]

Opinion:The manuscript addresses a fundamental and very interesting question in membrane biophysics. In this respect it is very suitable for publication in eLife. The problem is that it is unclear how trustworthy this large estimate is. The following suggestions may increase the credibility of the manuscript.

We appreciate the editor’s remarks in regard to the broad significance of the subject of our study, and are thankful to all reviewers for their suggestions and constructive criticisms. We trust that the new data and clarifications provided in the revised manuscript will address the reviewers’ concerns.

For the record, however, we believe it is important to clarify that our analysis of the extent of the membrane perturbations induced by Glt_Ph_ did not involve the use of a new enhanced sampling simulation method, as the Summary states. This component of the study, which has been substantially modified and expanded in the revision, entailed standard simulation methods, in different conditions and using both coarse-grained and all-atom forcefields – the latter being particularly challenging due to the very large size of the system. The new enhanced-sampling methodology mentioned in the Summary, now published in J Comp Chem, was used only to estimate the energetic cost of this perturbation. This second component has also been expanded in the revision.

Essential revisions:1) The in-plane distributions of the membrane deformations predicted by the simulations are inaccessible for experimental verifications. Moreover, the character of these deformations (types of strains in the case of 3D description or bending/stretching/chain tilting for a 2D description of the membrane) is not determined.

We understand that by in-plane distributions the reviewer refers to the two-dimensional representations of the deflection (depression/elevation) of the membrane mid-plane. We believe this kind of representation is important as it quantifies what is observed in the simulations, which in turn permits comparison of different conditions and conformational states. In order to facilitate a direct comparison with experiment, however, we also provided three-dimensional distributions (Figure 2, Figure 5, Video 1 and Video 2, Figure 8) of the lipid density around the protein – for both Glt_Ph_ and VcINDY. We believe that in time it will be possible to directly contrast this data with three-dimensional density maps derived from single-particle cryo-EM imaging of these transporters reconstituted in lipid nanodiscs – as has been possible for other membrane proteins.

With regard to the character of the deformation induced by all-inward Glt_Ph_, additional figures have been provided to underscore that the impact of the protein is primarily a change in the curvature of both membrane leaflets, as originally stated. Specifically, Figure 4—figure supplement 1 shows that the thickness of the membrane is, relatively speaking, influenced only minimally, both in terms of magnitude and range. Figure 4—figure supplement 2 reiterates the observation made in the original version that a significant change in alkyl-chain tilt (as quantified by the second-rank order parameter) is observed only in the inner leaflet, and only at the interface between the membrane and the scaffold. By contrast, the alkyl-chain tilt along the periphery of the transport domain, where the membrane is most significantly depressed, is essentially bulk-like. This data correlates with what can be inferred from the graphical 3D representations of lipid density in Figure 2, Figure 5, and Video 1and Video 2.

2) "In a recent breakthrough, we have developed and validated a promising free-energy simulation strategy to address this problem, which we refer to as Multi-Map (Fiorin, 2019)". The Fiorin paper is not in review here so we will not comment about it in detail. However, the draft that is included does not consolidate this bold statement. To our understanding it shows that the new formalism converges to the Helfrich-Canham theory and agrees with measurements at long distances (100Ang and more), as it should. However, we did not see any comparison regarding short distances which are relevant here. This statement should be removed.

We have no objection to rephrasing this statement to make it strictly factual: “In a recent development, we have reported a novel free-energy simulation strategy to address this problem, which we refer to as Multi-Map (Fiorin, 2019).”

We must note, however, that in this context the term “validation” refers to the fact we established the correctness of the enhanced-sampling algorithm, i.e., that it reproduces the expected thermodynamic ensemble, the target membrane morphology, and the mean forces along the biased collective variable from which the free energy is derived.

One way in which the proposed method was validated (as defined above) was by comparing it with unbiased-sampling simulations, for a problem that is fully quantifiable through both approaches (the free-energy of hydration of a hydrophobic cavity). In addition, the draft of Fiorin et al., enclosed with the original submission of the current article did include a comparison of calculated and measured bending moduli for small bilayers of different lipid types, in the presence and absence of cholesterol. Because experiments and computations probe very different length scales, what was evaluated is the relative change in this bending modulus upon addition of cholesterol. The results of this analysis were positive for the lipid types for which the experimental data is uncontroversial (POPC and DMPC). Thus, at least in regard to the effect of cholesterol on membrane rigidity of small lipid bilayers, the computational Multi-Map method performs no worse than existing experimental techniques.

3) The authors dismiss the Helfrich-Canham theory and its extensions, arguing that they do not hold for short distances. And yet, it would be insightful for the reader to know what values this approximate model and its extensions give for GltPh. At the very least it would indicate how high the 20 kcal/mol value is compared to existing theory.

We have not argued, here or elsewhere, that extensions to the Helfrich-Canham theory do not hold for small wave-length deformations, nor have we dismissed them. To the contrary, in Fiorin et al., we discuss how these extensions address the known limitations of the Helfrich-Canham model (described by us and others before us), in some cases quite elegantly. We have however argued that an alternative, direct PMF approach is advantageous as it does not entail important ad hoc assumptions (e.g. a bending modulus, a homogenous membrane, etc.) and avoids ill-defined methodological choices (as discussed below). In any case, it is beyond our specific expertise to carry out a systematic comparison of the multiple variations and extensions of the Helfrich-Canham functional that have been formulated over the years. As per *eLife*’s editorial policy, we believe this request is outside the scope of this revision.

Using our own simulation data, however, it is relatively straightforward to illustrate why an alternative to the Helfrich-Canham approach is advantageous. In the newly provided Figure 7—figure supplement 1A, we compare free-energy curves calculated with the direct PMF method (Figure 7) with 3 alternative results deduced from the Helfrich-Canham equation, for the same set of molecular configurations (i.e. for each of the umbrella-sampling simulations carried out to calculate the PMF). The two inputs for the calculation of the Helfrich-Canham energy are the assumed bending modulus (here we use *k*_c_ = 18 kcal/mol, standard for POPC) and the membrane curvature distribution across the bilayer plane, *c(x,y*). We evaluate this curvature distribution from analysis of the average mid-plane in from each umbrella-sampling simulation (18 μs per window) – see Materials and methods section. The Helfrich-Canham energy is then obtained by integrating the energy density [0.5 *k*_c_*c*^2^ (*x,y*)] over the area of the membrane. The three calculations in Figure 7—figure supplement 1A use the same value of the bending modulus, but differ in the level of resolution used to probe *c(x,y*). Linear regressions of the data in each case show that the Helfrich energy can be much larger, similar or somewhat smaller depending on the resolution with which the average local curvature of the membrane is evaluated. Needless to say, a smaller or larger value of the bending modulus might improve this correlation for a given resolution, but make it worse for another. Nevertheless, it seems clear that there is no reasonable choice of either *k*_c_ or the curvature-map resolution that would lead to Helfrich energies that are, say, 2-fold smaller than those we calculate with the Multi-Map method. The opposite is however true, i.e. Helfrich energies that are significantly larger than those deduced with a direct PMF method can be easily envisaged. In sum, going back to reviewers’ point, the 20 kcal/mol value reported in the manuscript is no larger than what would be reasonably inferred from non-microscopic theories of membrane energetics.

As illustrated in Figure 7—figure supplement 1B, a rigorous evaluation of the local membrane curvature *c(x,y*) is itself challenging and prone to large statistical errors despite extensive simulation time (18 μs per point, split in 1-μs fragments). Thus, when comparing RMS-curvatures across the bilayer obtained at different resolutions, the expected correlation is degraded at moderate to low curvature values. These ambiguities involved in extracting membrane curvature from molecular configurations further highlight the value of direct PMF calculations, of the kind now made possible by the Multi-Map method.

4) The derived energies of the membrane deformations, which could be indeed important for understanding the effects of the membrane lipid composition on the protein function, are not supported by any independent estimations and remain, therefore, completely dependent on the parameters of the computational model (forcefield parameters), and other details of the simulations. The authors should admit to it or provide experimental measurements.

The reviewer is right, evidently, that the results from any computer simulation or theoretical analysis are hypothetical until proven by experiment. This point has been clarified in the Discussion section.

5) No analysis is provided either of the effects of membrane tension, which may dramatically change the protein-mediated deformations and their distribution, or of the presence in the membrane of specific lipids such as cholesterol, PIPs, DAGs, etc, whose redistribution to the protein vicinity may moderate the elastic energy. This decreases the general impact of the results.

We appreciate the opportunity to discuss in detail the effect of membrane tension and membrane composition, as this is a recurrent question. As shown in the newly provided Figure 3, tensions of increasing magnitude do not dramatically change the nature of the deformations induced by all-inward Glt_Ph_. At the structural level, the effect of increased membrane tension is to gradually reduce the magnitude of the depression induced by the transport domains. Nevertheless, this perturbation is discernable even when the applied membrane tension is as high as 10 mN/m. It is important to note, however, that the energetic cost of the membrane deformation induced by Glt_Ph_ increases under membrane tension, as we show in the revised version of Figure 7B. Thus, while the amplitude of the perturbation caused by the transporter would be somewhat diminished if the membrane was under tension, the resulting energetic cost might be comparable to (or greater than) a condition where the membrane is not under tension.

With regard to the effect of lipid composition, we now compare 4 different conditions, namely POPC, POPE and 2:1 POPE-POPG at 298 K and DPPC at 323 K. Albeit simplified, we would argue these model membranes are not entirely unlike the conditions that have been probed in published liposome reconstitutions. As shown in Figure 4, the nature of the deformations induced by all-inward Glt_Ph_ is not dramatically changed by different lipid compositions either.

The notion that the functional regulation of membrane proteins by specific lipids (PIP2, cholesterol, etc) might be in part due to changes in the energetics of membrane plasticity is indeed an intriguing possibility, which we are currently pursuing for other systems where systematic experimental analyses have been conducted. We are unaware of such data for Glt_Ph_ and thus we believe that addressing this question is beyond the scope of this study.

6) Evidently, the authors are aware of the exceedingly high energy penalty due to lipid perturbation and try to explain it in Discussion. However, the arguments raised are not compelling. Thus, the tune of the manuscript should reflect the fact that all we have here is a computation that may or may not be correct.

See our response to points 4 and point 7.

7) Introduction, third paragraph: The question is nicely set between (1) "moving polar sidechains on their surface into the bilayer interior and exposing hydrophobic ones to the solvent" and (2) "the morphology of the lipid bilayer could adapt to the conformational state of the protein, and match the amino-acid make-up of the protein surface". The authors obviously come to the conclusion that the latter (2) is the case. However, at a considerable (20kcal/mol) cost. Given that the structure of all conformations are known, the authors should provide an estimate of the energetic cost for the rejected hypothesis (1).

We thank the reviewers for this suggestion, as this analysis enables us to quantify our claim that the cost of membrane bending is considerable smaller than the cost of the alternative (rejected) hypothesis.

This analysis is summarized in the newly provided Figure 6, for an equilibrated simulation snapshot of all-inward Glt_Ph_, and in Figure 6—figure supplement 1 for the crystal structure. The question that we pose is: what would be the hypothetical free-energy gain/cost of preserving a flat membrane when the transporter is in the all-inward state? To estimate this cost, we considered the two membrane configurations described in the figure – one deformed, as in our simulation, and a version thereof that has the same thickness but flat. (By ‘membrane’, we refer to the solvent-excluded region of the bilayer.) We then calculated the solvent-accessible surface area (SASA) for all protein residues in either case. The difference, i.e. ΔSASA, was then normalized by the SASA of each residue-type when fully solvent-exposed in a Gly-X-Gly peptide. The resulting ‘dehydration factor’ was then multiplied by the transfer free-energy of that residue type into the membrane, according to two independent hydrophobicity scales: one based on experimental measurements of the stability of WT and mutagenized OMPLA, published by Karen Fleming; and another published by Peter Tieleman, based on MD simulations.

Taken together, these results underscore how strikingly unfavorable it would be for the membrane to remain flat when the transport domains adopt the inward facing state. Small energy penalties from a large number of residues add up to very large totals – specifically about 30 kcal/mol for the Fleming scale and about 50 kcal/mol for the Tieleman scale. As originally stated, these penalties result from either exposing polar sidechains to the bilayer interior (R105, K230), or from exposing hydrophobic sidechains to the solvent; indeed, more than a dozen hydrophobic residues disfavor the flat membrane by 1 *k*_B_*T* or more, each. As one might expect, the polar residues that contribute the most are on the extracellular side of the transport domains, while the hydrophobic groups are on the intracellular side. A few hydrophobic residues on the extracellular side favor the flat membrane, but their total contribution is comparatively much smaller than the energetic penalty that opposes it.

It should be noted that, by definition, this ‘transfer free-energy’ examines the energetics of hydration/dehydration, neglecting the stabilizing effect of the hydrogen-bonding interactions between polar sidechains and the lipid headgroups described in Figure 5; many of these lipid contacts would be unfeasible if the membrane was flat, and would be replaced by water, penalizing this hypothetical state further.

In summary, the adaptation of the membrane to the conformational state of the transporter, while clearly costly, is nevertheless much more favorable than the alternative.

The simulations:8) The boundary conditions imposed on the membrane fragment are not specified. At the same time, in the absence of membrane tension, which seems to be the case here, the effects of the boundary conditions might be substantial.

The boundary conditions are now specified in the Materials and methods section, i.e., semi-isotropic pressure coupling and periodic boundary conditions. As mentioned, simulations are now presented with and without applied membrane tension. See Figure 3 and our response to point 5.

9) The method is based on combining coarse-grained and all-atom simulations, which can be tricky. In particular, for the all-atom simulations, the authors take the course-grained system they simulated and convert it to all-atom, while also (naturally) changing force-fields. Then they let the system relax and report that the trends observed are consistent with the course-grain simulations. It is theoretically possible that transitioning the system from the course-grained to the all-atom parameters is not that trivial, and maybe the fact that they start from an already equilibrated state introduces some bias. Could it be examined somehow?

This possibility was examined in the original submission, specifically in Figure 5—figure supplement 1AB (Formerly Figure 3—figure supplement 1). In this calculation, a perturbation of magnitude comparable to that induced by Glt_Ph_ is created in an all-atom lipid bilayer and allowed to relax over time. The perturbation dissipates well within 100 ns, which is shorter than the length of the Glt_Ph_ all-atom simulations.

In the revised version of the article, we further examine this question by quantifying the degree to which lipid molecules exchange between the first solvation shells around the protein and the rest of the membrane in the course of our 3 all-atom simulations of all-inward Glt_Ph_. As shown in the newly-provided Figure 5—figure supplement 1C, we find that, by the end of the simulation, about 30% of the lipids that are initially in these first solvation shells are replaced by other lipid molecules from further away from the protein – with the overall lipid number in each shell approximately constant. These exchanges would permit a change in membrane morphology, if the such change was favorable. The fact that the membrane remains deformed, despite the results from these two control calculations, suggests that this deformation is the preferred state of the molecular system, and not an artifact of the coarse-grained forcefield, preserved at the all-atom level.

10) It is unfortunate that DPPC (transition temperature 41C) was chosen for many simulations. Why? Also, simulations using POPC (transition temperature -2C) are shown. The fact that the results are quasi identical seems rather worrying than comforting. Why these choices? Why not using a mixture, it is not unthinkable that in a mixture the protein would recruit specific lipids on these interfaces between transporter and scaffold domains. Please comment. Or, better, examine in simulations.

Although we did not observe a phase transition in any of our original simulations, we recognize that the use of DPPC near the (coarse-grained) transition temperature is a concern. To address this concern, we have repeated all the coarse-grained simulations using different conditions. Specifically, we repeated the triplicated simulations of all-outward, one-inward, two-inward, and all-inward Glt_Ph_ using POPC at 298 K, and show the results in Figure 2. We also carried out triplicated simulations of all-inward Glt_Ph_ in POPE and in a 2:1 POPE:POPG mixture at 298 K, as well as in DPPC at 323 K, i.e., well above the transition temperature. The results of these simulations are shown in Figure 4. The triplicated simulations of VcINDY, in all-outward and all-inward conformations were also repeated using POPC at 298 K, and the results shown in Figure 8. Taken together, these new simulation data convey that the effects we describe are not critically dependent on variations in the lipid composition of the membrane. Please see also our response to point 5.

Presentation:11) Introduction: "These perturbations develop to accommodate the amino-acid composition and specific structural features of the protein surface.": This deserves a reference to the Piezo channel, the only protein that displays clear structural features that force the membrane into bending, and for which a theory and experiments about how membrane bending (and flattening / the physics of the membrane) is exploited to gate the channel was presented.

A reference has been added.

12) The general reader might not know what the second-rank order parameter of the lipid alkyl chains is.

An explanation has been added (see caption of Figure 4—figure supplement 2).

13) Figure 2, Figure 3 and associated, in the context of "Transport domains bend the membrane, while scaffold domains anchor it": While we understand the measurements, the results and the importance of all these results, the presentation is unclear with respect to the setting of the 0-level. From the captions "2(A) The deflection is quantified by calculating the mean value of the Z coordinate of the bilayer across the X-Y plane." and "(A) Deflection of the membrane mid-plane relative to a flat surface". For example, the image in 3a), all the membrane has negative deformation. Wouldn't that correspond to creation of a net elastic and potential energy? Shouldn't the bilayer as a whole still go towards flat and as a result one has negative deformation next to the transporter domains and positive deformation next to the scaffold domain. This does not change anything to the results in terms of local deformation, right? Is the bilayer held in place at the periphery of the simulation box?

We have clarified this point in the revised version.

The Z-coordinate of the membrane mid-plane is obtained from an average over all snapshots collected in a given simulation. The protein starts perfectly centered in the membrane, but in the course of a simulation it tumbles and rotates freely (and the membrane, which is initially flat, adapts accordingly). To analyze the simulation data, therefore, all snapshots must be first transformed so that the center-of-mass and orientation of the Glt_Ph_ trimer are consistent across the data set. To do so, all atomic coordinates (protein, membrane, solvent) are roto-translated equally, in a manner that results in an optimal (least-squares) fitting of the protein backbone. Note that this transformation does not alter in any way the relative coordinates of the atoms in the molecular system. Through this procedure, the membrane shape is defined with good precision in the vicinity of the protein. Conversely, at the edges of the simulated membrane patch, this procedure leads to large “apparent” fluctuations and an ill-defined average. Nevertheless, with extensive sampling it becomes possible to define a 0-level with confidence, i.e. a broad region far from the protein where the membrane is flat on average, and where the “apparent” fluctuations induced by the analysis process are small. In our case, that 0-level is about ~200 Å from the protein center.

The original version of Figure 5A (formerly 3A) was a close-up view of the membrane close to the protein – hence this 0-level wasn’t clear. To avoid confusion, in the new version we zoom out and present the data exactly as for the coarse-grained simulations in Figure 2. As mentioned, the 0-level becomes clear at about 200 Å from the protein center.

The membrane is not held in place, anywhere along the periphery of the system, or elsewhere. The perturbation caused by the transporter does however decay naturally away from the protein surface and so the membrane does indeed become flat, as it should. That is the region where the 0-level is set.

14) Further on this: Why is the scaffold domain considered the 'anchor'? Why does the membrane next to the transporter domains (that are in the inward (down) orientation) move down, rather than the scaffold domain moving upwards? Overall – after sufficiently long relaxation of the all-inward structure – shouldn't the overall level of the membrane remain 0 and the membrane next to the transporter domains shift downwards, but the membrane next to the scaffold domains shift upwards? Especially, in light of Figure 3C, which seems to show that there are many more key amino acids in the transporter than in the scaffold domain, one would expect that the transport domain dominates the relative motion. It is unclear how in panels like Figure 2A left, 2A right, Figure 3A, all membrane deformation values can be positive or negative? Is this the reason why such huge energy penalties result from the analysis?

As explained above, the 0-level can be defined reliably at about 200 Å from the center of the protein, which is about 150 Å from its surface. Relative to this 0-level, the change in the membrane mid-plane is much smaller at the interface with the scaffold domain than at the interface with the transport domains (see 1D profiles in Figure 2D, formerly Figure 3C). The relative change in this mid-plane when comparing all-outward to all-inward is also much larger for the transport domains. Therefore, it seems logical to think of the scaffold domain as the ‘anchor’. As noted, however, the scaffold isn’t entirely stationary. Comparing the all-outward with the all-inward state, we detect a small net displacement inwards and the transport domains move inwards too. However, we certainly do not observe an upwards displacement, relative to the 0-level. [Of course, one could arbitrarily redefine the 0-level so that the membrane mid-plane appears unperturbed at the transport domain, but that 0-level would not be consistent with the lipid bilayer further from the protein.]

Ultimately, the geometry of the membrane and the position of the protein therein is determined by a complex free-energy function that, as we argue, includes the conformational preference of the membrane itself and not only a protein-centric perspective. The geometry described by the reviewer for all-inward Glt_Ph_ seems entirely plausible a priori (i.e., the membrane is elevated near the scaffold domain and approximately balances out the opposite perturbation caused the transport domains), but in our view is not more intuitive than other alternatives. Arguably, the purpose of molecular simulations such as those presented in our article is precisely to investigate complex, multi-component problems such as this – notwithstanding the caveat in point 4.

15) Discussion section: The authors find a large energetic penalty for the inward facing state due to membrane deformation. They note that the smFRET studies displayed almost zero energy difference between the states. In contrast the HS-AFM study revealed that the inward facing state indeed was the high energy state (like here). The authors mention that the HS-AFM study was performed in rather densely packed membrane where no such long range lipid relaxations are possible as in the study here, yet it is a flat membrane. In this context, it is also noticeable that the smFRET studies are performed either in detergent or on transporters in small vesicles that are tethered in outside-out configuration only – which goes against the bowl-shaped structure of GltPh – of which the authors see the preference for membrane bending in Figure 1 (side view), which might favor the adoption of the inward-facing state in the smFRET experiments.

We thank the reviewers for pointing out this interesting possibility, which has been noted in the Discussion section.